# EHRNoteQA: An LLM Benchmark for Real-World Clinical Practice Using Discharge Summaries

**Sunjun Kweon**[1][*]**, Jiyoun Kim**[1][*]**, Heeyoung Kwak**[2,3]**, Dongchul Cha**[2,4]**,**
**Hangyul Yoon**[1]**, Kwanghyun Kim**[5]**, Jeewon Yang**[1]**, Seunghyun Won**[6]**, Edward Choi**[1,†]
KAIST[1]   NAVER Digital Healthcare LAB[2]   Naver Cloud[3]
NAVER Healthcare LAB[4] Ewha Womans University College of Medicine[5]
Seoul National University Bundang Hospital[6]
{sean0042,jiyoun.kim,edwardchoi}@kaist.ac.kr[1]

## Abstract

Discharge summaries in Electronic Health Records (EHRs) are crucial for clinical decision-making, but their length and complexity make information extraction challenging, especially when dealing with accumulated summaries across multiple patient admissions. Large Language Models (LLMs) show promise in addressing this challenge by efficiently analyzing vast and complex data. Existing benchmarks, however, fall short in properly evaluating LLMs' capabilities in this context, as they typically focus on single-note information or limited topics, failing to reflect the real-world inquiries required by clinicians. To bridge this gap, we introduce **EHRNoteQA**, a novel benchmark built on the MIMIC-IV EHR, comprising 962 different QA pairs each linked to distinct patients' discharge summaries. Every QA pair is initially generated using GPT-4 and then manually reviewed and refined by three clinicians to ensure clinical relevance. EHRNoteQA includes questions that require information across multiple discharge summaries and covers ten diverse topics, mirroring the complexity and diversity of real clinical inquiries. We offer EHRNoteQA in two formats: open-ended and multi-choice question answering, and propose a reliable evaluation method for each. We evaluate 27 LLMs using EHRNoteQA and examine various factors affecting the model performance (*e.g., the length and number of discharge summaries*). Furthermore, to validate EHRNoteQA as a reliable proxy for expert evaluations in clinical practice, we measure the correlation between the LLM performance on EHRNoteQA, and the LLM performance manually evaluated by clinicians. Results show that LLM performance on EHRNoteQA have higher correlation with clinician-evaluated performance (Spearman($\rho$): 0.78, Kendall($\tau$): 0.62) compared to other benchmarks, demonstrating its practical relevance in evaluating LLMs in clinical settings. EHRNoteQA is publicly available under PhysioNet credential access at https://doi.org/10.13026/acga-ht95, and the code is available at https://github.com/ji-youn-kim/EHRNoteQA.

## 1  Introduction

Discharge summaries are clinical notes in Electronic Health Records (EHRs) written by healthcare professionals upon patient discharge, encapsulating a patient's overall medical course from admission to discharge. These documents provide crucial patient information such as medical events, diagnoses, and treatments, which are vital for healthcare professionals in making informed clinical decisions

---

[*]These authors contributed equally

[†]Corresponding author

[13], particularly in scenarios such as patient readmission and transitions of care (i.e., patient handoffs) [5, 7]. Healthcare professionals often need to refer to and compare information across multiple discharge summaries to gain a comprehensive understanding of the patient's medical history, especially when a patient is admitted multiple times. Nevertheless, the length and complexity of discharge summaries often make it challenging to extract necessary information efficiently [14], a problem exacerbated when dealing with accumulated discharge summaries across multiple patient admissions.

Recent advancements in Large Language Models (LLMs) present a promising solution to this challenge. With their capability to efficiently analyze extensive and complex information within EHRs [3, 16, 18, 47, 63, 67], LLMs have the potential to serve as question-answering (QA) agents within healthcare institutions, assisting healthcare professionals in obtaining answers to queries regarding patient discharge summaries. However, prior to deploying LLMs in such practical scenarios, it is necessary to develop a benchmark to assess the performance of LLMs in this specific context.

While several clinical benchmarks [23, 43, 24, 19] have been recently employed to evaluate LLMs in the clinical domain [50, 51, 40, 33, 55, 41, 8], they primarily focus on general medical questions (*e.g., "What is the treatment for adenomyosis?"*), without addressing practical use cases involving specific patient records. Furthermore, although QA datasets based on EHR discharge summaries exist [44, 65, 15, 38], they fall short in reflecting the diverse real-world scenarios in which physicians inquire on a patient's discharge summary. These are the following reasons: 1) the datasets only consist of questions that require information within a single note (*e.g., "What is the dosage of Nitroglycerin?"*), lacking questions that necessitate information across multiple notes (*e.g., "How did the dosage of Nitroglycerin change between visit x and y?"*) 2) most of the datasets are constructed based on predefined clinical annotations (e.g., i2b2 [58]) or with a predetermined topic focus (e.g., relations, drugs), resulting in a constrained scope of question topics. These limitations underscore the need for a new benchmark to effectively evaluate LLMs in answering queries posed by healthcare professionals in actual clinical practice.

To this end, we present **EHRNoteQA**, a novel benchmark to evaluate LLMs in real-world clinical scenarios for answering clinicians' questions regarding patient discharge summaries. EHRNoteQA is built upon the MIMIC-IV EHR [26], and consists of 962 different QA pairs each linked to distinct patients' discharge summaries. Each QA pair is initially generated using GPT-4 and then carefully reviewed and refined individually by three clinicians to ensure clinical relevance. EHRNoteQA reflects real-world clinician queries for the following reasons: 1) it includes questions that require information across multiple discharge summaries (*e.g., "How did the post-discharge medication regimen change between visits x and y?"*), 2) the questions span a diverse set of 10 topics (*i.e., treatment, assessment, problem, etiology, sign/symptom, vitals, test results, history, instruction, plan*). For each question, EHRNoteQA includes a single complete answer along with four distractor answer choices, enabling evaluation of LLMs in both open-ended and multi-choice formats. For both formats, we provide a reliable evaluation method for measuring LLM performance.

We evaluate 27 LLMs using EHRNoteQA and analyze various factors that influence model performance, such as the length and number of discharge summaries. Furthermore, to ensure practical relevance in real clinical scenarios, we conduct additional experiments to validate EHRNoteQA as a reliable proxy for actual expert evaluation; we first have the clinicians manually evaluate the LLM performance on discharge summary-based questions asked by unknown clinicians (to avoid any bias). Then we measure the correlation between this performance, and automatically-evaluated LLM performance on diverse benchmarks including EHRNoteQA. The results show that LLM scores on EHRNoteQA have a higher correlation with clinician-evaluated LLM scores (Average: Spearman($\rho$): 0.78, Kendall($\tau$): 0.62) compared to other benchmarks, highlighting its practical relevance in evaluating LLMs in real clinical settings.

## 2 Related Works

**General Clinical Benchmark.** Recent studies [50, 51, 40, 33, 55, 41, 8] have employed various benchmarks to assess the performance of LLMs in the clinical domain. Benchmarks such as MedQA [23] and MedMCQA [43] consist of questions sourced from medical licensing exams. PubMedQA [24] comprises QA pairs formulated from PubMed articles, with questions derived from their titles. Furthermore, MMLU [19] is a multi-task test set of various topics, including clinical subjects such as

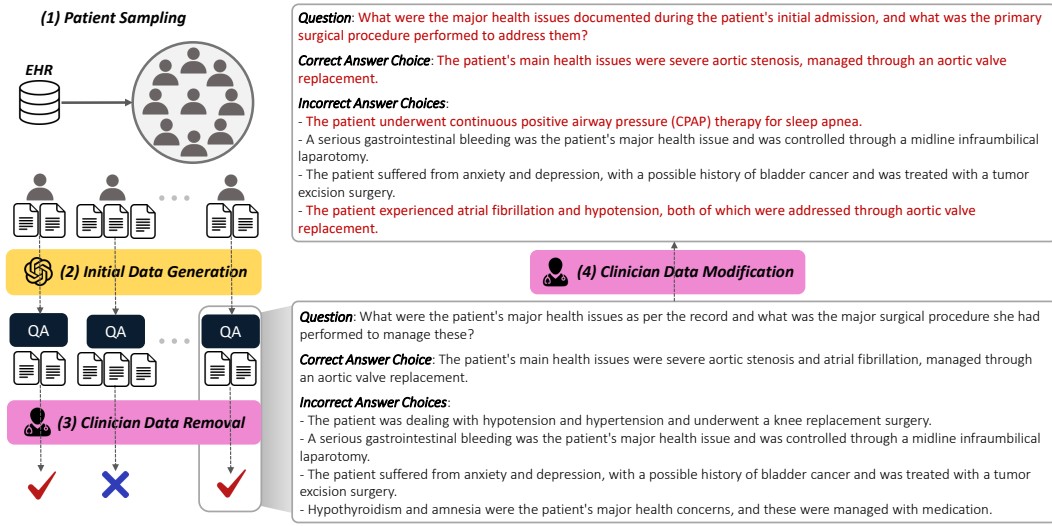

Figure 1: Overview of the EHRNoteQA dataset construction process. It involves (1) patient sampling from the MIMIC-IV EHR database, (2) initial data generation using GPT-4, (3) clinician removal of improper data, and (4) clinician modification of the data.

medical genetics and professional medicine, sourced from textbooks and standardized tests. While these benchmarks may be useful for assessing the clinical domain knowledge of LLMs, they are not sufficient for evaluating the practical capabilities of models in assisting healthcare professionals in real-world clinical settings [37]. Specifically, tasks such as extracting and interpreting information from individual patient records are not included in these benchmarks. Therefore, achieving high performance on these benchmarks does not ensure that a model will perform well in these realistic scenarios unless explicitly evaluated in such settings.

Table 1: Comparison of EHRNoteQA with existing discharge summary QA datasets.

| Dataset | Questions | Documents | Patients | Answer Format | Grounded Documents | Topics |
|---------|-----------|-----------|----------|---------------|-------------------|--------|
| Pampari et al. [44] | 73,111 | 303 | 303 | Text Span | Single | 5 |
| Fan [15] | 245 | 138 | 138 | Text Span | Single | 1 |
| Yue et al. [65] | 1,287 | 36 | 36 | Text Span | Single | 5 |
| Moon et al. [38] | 96,939 | 505 | 505 | Text Span | Single | 1 |
| EHRNoteQA (Ours) | 962 | 1,659 | 962 | Multi-Choice & Open-Ended | Multiple | 10 |

**Discharge Summary QA Datasets.** Previous works have explored patient-specific QA using EHR discharge summaries (see Table 1 for comparison). Pampari et al. [44] introduced emrQA, the first public clinical note QA dataset generated from expert-annotated question templates and i2b2 annotations [58, 57, 59, 60]. Yue et al. [65] proposed a dataset built on MIMIC-III discharge summaries [25] by extracting answer evidence, followed by generating corresponding questions using a neural question generator model. Additionally, Fan [15] and Moon et al. [38] proposed discharge summary-based QA datasets on why-questions and drug-reason relations, respectively.

However, these existing datasets fall short in reflecting the complexity and diversity of real-world questions posed by physicians. Firstly, they only address questions based on a single note, neglecting the frequent clinical need to reference multiple discharge summaries for patients with multiple admissions. Secondly, the question topics are constrained. For example, emrQA is confined to topics within i2b2 annotations, such as smoking, medication, obesity, and heart disease. Although Yue et al. [65] aimed to increase diversity, Lehman et al. [30] noted that 96% of the questions still follow emrQA's templates, indicating continued limitation. Fan [15] focused solely on why-questions, limiting the scope of topics, while Moon et al. [38] centered on drug-reasoning based on n2c2 annotations [20].

Table 2: Number of patients and average discharge summary length for MIMIC-IV, sampled MIMIC-IV, and EHRNoteQA across different levels and number of admissions (# *D.S.*). *D.S.* denotes Discharge Summaries.

| Category | | MIMIC-IV | | Sampled | | EHRNoteQA | |
| --- | --- | --- | --- | --- | --- | --- | --- |
| Level | # D.S. | # Patients | Avg. Length | # Patients | Avg. Length | Patients | Avg. Length |
| 1 | 1 | 38,926 | 1,819 | 275 | 1,787 | 264 | 1,812 |
| | 2 | 437 | 2,147 | 275 | 2,146 | 265 | 2,085 |
| 2 | 1 | 44,645 | 3,514 | 150 | 3,501 | 145 | 3,497 |
| | 2 | 14,176 | 4,470 | 150 | 4,581 | 144 | 4,520 |
| | 3 | 1,161 | 4,956 | 150 | 5,030 | 144 | 5,102 |
| Total | | 99,345 | - | 1,000 | - | 962 | - |

# 3 EHRNoteQA

## 3.1 Patient Filtering & Sampling

We construct EHRNoteQA (see Figure 1) using discharge summaries from the MIMIC-IV EHR database [26], containing anonymized patient records from Beth Israel Deaconess Medical Center (BIDMC) between 2008 and 2019. The MIMIC-IV database includes a total of 331,794 discharge summaries for 145,915 unique patients, averaging 2.3 notes (admissions) per patient. However, these discharge summaries are often lengthy, with the average length of a single patient's accumulated discharge summaries around 8,000 tokens. This poses a challenge for current LLMs, as few can process contexts longer than 8,000 tokens.

To address this, we first preprocess the notes by removing excessive whitespace, reducing the average length by 10% (see Appendix C.1). We then categorize patients by the length of their accumulated discharge summaries to match the context-length of existing LLMs [3]. Specifically, we divide the patients into two groups: Level 1, consisting of patients whose accumulated discharge summary length does not exceed 3,000 tokens, suitable for models that can handle up to 4,000 tokens, and Level 2, consisting of patients whose accumulated discharge summary length ranges from 3,000 to 7,000 tokens, suitable for models that can handle up to 8,000 tokens. Each group includes a 1,000-token buffer to accommodate additional prompts and outputs beyond the discharge summary text. Together, these two groups cover about 70% of the patients in the MIMIC-IV database.

As shown in Table 2, the first group (Level 1) includes patients who had been admitted either one or two times, as indicated by the number of discharge summaries. The second group (Level 2) includes patients who had been admitted one to three times. We randomly sample a total of 1,000 patients—550 from Level 1 and 450 from Level 2—and use their discharge summaries for the next step.

## 3.2 Initial Data Generation

Based on the discharge summaries sampled from Section 3.1, we utilize GPT-4 [42] to generate the initial draft of EHRNoteQA. For each patient's accumulated discharge summaries, GPT-4 is tasked to generate the initial QA data, including 1) a clinically meaningful question that clinicians might ask regarding the patient's discharge summaries, 2) the complete answer to the question, and 3) four incorrect answer options (see Figure 1 bottom right for an example). This data construction enables EHRNoteQA to evaluate LLMs in two ways: through an *open-ended method* in which the model's free-form answer to the question is evaluated, and through a *multi-choice method* in which the selected answer choice is evaluated. Note that we use GPT-4 on Azure's HIPAA-compliant platform (see Appendix B.2), adhering to the MIMIC-IV data usage regulations. Detailed prompts and costs can be found in the Appendix C.2.

---

[3]During this study, most open-source LLMs support up to 4,000 tokens, with some handling up to 8,000. We exclude LLMs with a 2,000-token context length as they cannot even take a single discharge summary.

Table 3: Examples and proportions of question categories in EHRNoteQA. Note that the proportions do not sum to 100% since a single question can be categorized into multiple categories when containing more than one topic (*e.g., Etiology and Treatment: "What were the primary causes of the patient's shortness of breath and how were these managed during the hospital stay?"*)

| Category | Example | Proportion |
|---|---|---|
| Treatment | What was the treatment provided for the patient's left breast cellulitis? | 64% |
| Assessment | Was the Mitral valve repair carried out successfully? | 19% |
| Problem | What was the main problem of the patient? | 19% |
| Etiology | Why did the patient's creatinine level rise significantly upon admission? | 20% |
| Sign/Symptom | What was the presenting symptom of the patient's myocardial infarction? | 12% |
| Vitals | What was the range of the patient's blood pressure during second stay? | 3% |
| Test Results | What were the abnormalities observed in the patient's CT scans? | 14% |
| History | Has the patient experienced any surgical interventions prior to the acute appendicitis? | 12% |
| Instruction | How was the patient instructed on weight-bearing after his knee replacement? | 3% |
| Plan | What is the future course of action planned for patient's left subclavian stenosis? | 5% |

## 3.3 Clinician Review & Modification

Each question-answer pair generated by GPT-4 is reviewed and modified by three clinicians to ensure clinical relevance. For each QA pair, the clinicians are provided with the discharge summaries, the question, the complete answer, and the four incorrect answers. They follow the instructions below for the review and modification process.

Firstly, the clinicians assess whether the GPT-4-generated questions are appropriate. Questions that are not clinically important based on the patient's discharge summaries (*e.g., "What were the patient's symptoms prior to being diagnosed with a wound abscess?"*) or are not in a form that a physician would typically ask (*e.g., "Can we conclude that the patient presented any positive leukocyte or nitrate urine test result?"*) are identified and removed. Of the initial 1,000 questions, 38 were removed, leaving 962.

Next, the data that passed the initial review undergoes three further revisions. Firstly, questions that are ambiguous, overly detailed, or asks for unnecessary additional detail are modified. Of the 962 questions, 206 underwent this revision process. Subsequently, based on the revised questions, the answers are also reviewed and modified to align with the modified question. Even if the questions are not modified, answers that are inaccurate or incomplete are also revised. In total, 338 answers were modified out of the 962. Finally, the incorrect answer choices are revised to represent plausible distractors instead of being obviously incorrect (e.g., clinically contradictory or not mentioned in the discharge summary, making them too easy to identify as incorrect). Among the 3,842 incorrect answers (962 questions with 4 incorrect answers each), 966 were revised. Figure 1 shows an example of these revisions, with more examples provided in the Appendix C.3.

## 3.4 Data Analysis

Our final EHRNoteQA dataset, after revisions by clinicians, comprises 962 questions paired with the discharge summaries of 962 distinct patients. As shown in Table 2, Level 1 data includes a total of 529 patients: 264 patients admitted once (one discharge summary) and 265 patients admitted twice. Level 2 data includes a total of 433 instances: 145 patients admitted once, 144 patients admitted twice, and 144 patients admitted three times.

We analyze the types of information (topics) addressed by the questions in EHRNoteQA. We establish 10 categories (*i.e., treatment, assessment, problem, etiology, sign/symptom, vitals, test results, history, instruction, plan*) and define the criteria for each category (*e.g., Etiology: Questions asking about the causes of a patient's symptoms and problems*). Each of the 962 questions were categorized manually by the authors, with examples and proportions presented in Table 3. The categorization criteria along with additional examples can be found in the Appendix D.

# 4 Experiments

## 4.1 Setup

**Models.** We evaluate the performance of 27 LLMs on EHRNoteQA including 3 GPT series (*i.e.,* GPT-4, GPT-4-turbo, GPT-3.5-turbo) and 24 open-source LLMs (refer to Table 4). These LLMs are instruction-tuned models capable of processing context lengths of at least 4k tokens. For our Level 1 data, we evaluate LLMs with maximum context length of at least 4k tokens, while for Level 2 data, we evaluate LLMs capable of handling at least 8k tokens. To ensure privacy preservation during the evaluation process, we leverage Azure's HIPAA-compliant platform for the GPT series models, while performing local inference for the open-source LLMs (see Appendix B.2 for detailed settings).

**Evaluation.** As stated in Section 3.2, EHRNoteQA supports both open-ended and multi-choice QA. For open-ended QA, we use GPT-4 to evaluate the model outputs instead of using traditional methods such BLEU [45] or ROUGE-L [34], which often exhibit low correlation with human evaluations [36, 16]. Specifically, we prompt GPT-4 to evaluate the model output as either correct or incorrect, given the patient discharge summaries, the question, the correct answer, and the model output. To validate the accuracy of this evaluation method, three clinicians manually scored 10 different question-output pairs for each of the 27 models (totaling 270 pairs) and the results were compared to the evaluations by GPT-4 on the same set. The resulting Cohen's Kappa agreement ($\alpha$) between evaluations by GPT-4 and the three clinicians are 0.757, 0.855, and 0.880, respectively, while the intra-agreement among clinicians range from 0.808 to 0.903. Although the agreement of 0.757 falls slightly below this range (0.808 to 0.903), the overall high agreements (0.855 and 0.880) between GPT-4 and the clinicians, as well as the intra-clinician agreement, demonstrate the reliability of this open-ended evaluation method using GPT-4.

For multi-choice evaluation, we also use GPT-4 instead of the widely used probability-based evaluation methods [17, 32], which will be further discussed in Section 4.3. Specifically, we input the model output, the five answer choices, and the correct answer, and prompt GPT-4 to evaluate the model output as either correct or incorrect. The authors manually assessed this evaluation method[4], which revealed a 98% accuracy. In summary, we use GPT-4 evaluation for both open-ended and multi-choice QA on EHRNoteQA as our primary evaluation method. Detailed information about evaluation such as GPT-4 prompt, pricing, and reliability check is provided in Appendix E.

## 4.2 Results

The results of evaluating 27 LLMs on EHRNoteQA using both multi-choice and open-ended answering formats are presented in Table 4. The following analyses can be derived from the results:

**Impact of Model Size.** Smaller models generally tend to score lower compared to larger models. However, exceptions exist based on factors such as the foundation model and the fine-tuned instruction dataset. For instance, the 7b size model (*Mistral-7B-OpenOrca*) outperforms a model ten times its size (*Llama2-70b-chat*). Furthermore, while GPT-4 achieves the highest scores in both multi-choice and free-text evaluations, *Llama3-70b-Instruct* shows performance close to GPT-4.

**Impact of Foundation Model.** The choice of the foundation model significantly affects performance. For instance, when comparing model scores of the same size, those based on *Mistral-7b* (average: 73.29) generally outperform those based on *Llama2-7b* (average: 65.69).

**Impact of Instruction Set.** Performance varies significantly depending on the instruction set used to train the model, even with the same foundation model. For example, the instruction-tuned models based on *Llama-13b* exhibit a wide range of scores: 71.46 to 86.01 for multi-choice format and 66.16 to 79.21 for open-ended format.

**Effectiveness of Clinical Instruction Sets.** Even when models are trained on clinical domain instructions, their performance varies depending on the specific task. For example, *qCammel-13* and *qCammel-70* trained on synthetic patient-doctor dialogues, do not show significant advantages compared to those trained on general instructions, whereas *Asclepius-7* and *Asclepius-13*, trained on synthetic clinical note instructions, generally score higher on EHRNoteQA.

---

[4]Unlike open-ended, multi-choice format do not require clinician involvement for reliability check, as we only need to check if the model's output (*e.g., "The answer is A, because..."*) contains the correct answer index.

Table 4: Results of 27 LLMs using both multi-choice and open-ended question answering methods for EHRNoteQA. Empty cells in Level 2 indicate that the model does not support context lengths up to 8k. [1]Asclepius is trained to provide only open-ended responses.

| Size | Model | Multi-Choice | | Open-Ended | | Foundation | Reference |
|------|-------|:----------:|:----------:|:----------:|:----------:|------------|-----------|
| | | Level 1 | Level 2 | Level 1 | Level 2 | | |
| | GPT4 | 97.16 | 95.15 | 91.30 | 89.61 | | [42] |
| | GPT4-Turbo | 95.27 | 94.23 | 91.30 | 86.61 | | [42] |
| | GPT3.5-Turbo | 88.28 | 84.99 | 82.23 | 75.52 | | [6] |
| | Llama3-70b-Instruct | 94.33 | 91.92 | 89.04 | 86.84 | Llama3-70b | [1] |
| | Llama2-70b-chat | 84.88 | – | 78.83 | – | Llama2-70b | [56] |
| 70B | qCammel-70 | 85.63 | – | 78.26 | – | Llama2-70b | [55] |
| | Camel-Platypus2-70b | 89.79 | – | 78.83 | – | Llama2-70b | [28] |
| | Platypus2-70b-Instruct | 90.36 | – | 80.53 | – | Llama2-70b | [28] |
| 8x7B | Mixtral-8x7b-Instruct | 87.52 | 86.61 | 88.28 | 81.52 | Mistral-7b | [22] |
| 30B | MPT-30b-Instruct | 79.96 | 75.52 | 67.11 | 62.59 | MPT-30b-8k | [39] |
| | Llama2-13b-chat | 73.65 | – | 70.32 | – | Llama2-13b | [56] |
| | Vicuna-13b | 82.04 | – | 70.51 | – | Llama2-13b | [9] |
| | WizardLM-13b | 80.91 | – | 74.67 | – | Llama2-13b | [62] |
| 13B | qCammel-13 | 71.46 | – | 66.16 | – | Llama2-13b | [55] |
| | OpenOrca-Platypus2-13b | 86.01 | – | 79.21 | – | Llama2-13b | [29] |
| | Camel-Platypus2-13b | 78.07 | – | 67.86 | – | Llama2-13b | [28] |
| | Synthia-13b | 79.21 | – | 74.48 | – | Llama2-13b | [54] |
| | Asclepius-13b[1] | – | – | 75.24 | – | Llama2-13b | [27] |
| | Gemma-7b-it | 77.50 | 67.21 | 63.71 | 54.27 | Gemma-7b | [52] |
| | MPT-7b-8k-instruct | 59.55 | 51.27 | 56.71 | 53.81 | MPT-7b-8k | [39] |
| | Mistral-7b-Instruct | 82.04 | 64.90 | 72.97 | 53.81 | Mistral-7b | [21] |
| | Dolphin-2.0-mistral-7b | 76.18 | – | 69.75 | – | Mistral-7b | [12] |
| 7B | Mistral-7b-OpenOrca | 87.15 | – | 79.58 | – | Mistral-7b | [31] |
| | SynthIA-7b | 78.45 | – | 74.67 | – | Mistral-7b | [53] |
| | Llama2-7b-chat | 65.78 | – | 58.98 | – | Llama2-7b | [56] |
| | Vicuna-7b | 78.26 | – | 59.74 | – | Llama2-7b | [9] |
| | Asclepius-7b[1] | – | – | 66.92 | – | Llama2-7b | [27] |

**Impact of Note Length on Model Performance.** When comparing the scores of models capable of handling both Level 1 and Level 2 data (models supporting at least 8k context length), we observe that scores generally decrease from Level 1 to Level 2 within the same model. This decrease highlights the increased complexity in comprehending and interpreting extensive clinical contexts presented in longer patient notes. Notably, while *Mixtral-8X7B* shows a minor score reduction, *Mistral-7B* exhibits a more significant drop. This reveals an important insight: while both models perform similarly at Level 1, they diverge significantly at Level 2, indicating that a model's proficiency at Level 1 does not necessarily translate to similar performance at Level 2, where tasks involve longer patient notes.

**Impact of Number of Notes on Model Performance.** In addition to comparing model performance across different note lengths, we also evaluate the impact of the number of notes on model performance. Specifically, within the same level data, we compare the scores of data with only one note to those of data with multiple notes (e.g., 2, 3 respectively). The results are presented in Appendix E.2 Consistent with the observed performance regarding note length, scores generally decrease as the number of notes increase for the same model, underscoring the challenge in interpreting cumulative discharge summaries across multiple admissions. Notably, while the scores for *Camel-Platypus2-13b* and *Llama2-13b-chat* are similar for the one-note questions in Level 1, *Camel-Platypus2-13b* shows a minor 5.21% reduction from one to two notes, whereas *Llama2-13b-chat* exhibits a more significant 13.5% drop. This reveals that proficiency with a single note does not necessarily translate to similar performance with two notes, where tasks involve interpreting information across more patient notes.

## 4.3 The Reliability of EHRNoteQA as a Proxy for Clinician Evaluations

Prior to deploying an LLM as a QA agent in clinical settings such as responding to questions on discharge summaries, it is desirable for clinicians to directly assess the models' performance [46]. A

Table 5: The correlation between *clinician-evaluated LLM scores* by three individual clinicians and 1) the evaluations by other clinicians, 2) EHRNoteQA, 3) various other benchmarks, and 4) the methods of evaluating the open-ended and multiple-choice formats of EHRNoteQA. Within the benchmark comparison, bold indicates the highest correlation and underlined the second highest.

| | | Clinician A | | Clinician B | | Clinician C | |
|---|---|---|---|---|---|---|---|
| | | Spearman($\rho$) | Kendall($\tau$) | Spearman($\rho$) | Kendall($\tau$) | Spearman($\rho$) | Kendall($\tau$) |
| **Intra-Clinician correlation** | | | | | | | |
| | **Clinician A** | - | - | 0.854 | 0.712 | 0.947 | 0.834 |
| | **Clinician B** | 0.854 | 0.712 | - | - | 0.867 | 0.724 |
| | **Clinician C** | 0.947 | 0.834 | 0.867 | 0.724 | - | - |
| **Benchmark Comparison** | | | | | | | |
| **EHRNoteQA** | **Open-Ended** | **0.770** | 0.609 | **0.805** | **0.617** | 0.801 | 0.657 |
| | **Multi-Choice** | 0.766 | **0.661** | 0.732 | 0.574 | **0.812** | **0.661** |
| **Discharge Summary QA** | emrQA | 0.696 | 0.522 | 0.653 | 0.518 | 0.661 | 0.475 |
| | Yue et al. | 0.509 | 0.344 | 0.502 | 0.315 | 0.542 | 0.344 |
| **Clinical Benchmark** | MedQA | 0.590 | 0.453 | 0.497 | 0.354 | 0.683 | 0.535 |
| | MedMCQA | 0.672 | 0.512 | 0.505 | 0.378 | 0.737 | 0.594 |
| | PubMedQA | 0.122 | 0.100 | 0.071 | 0.059 | 0.167 | 0.088 |
| | MMLU* | 0.684 | 0.543 | 0.646 | 0.503 | 0.804 | 0.637 |
| **General Benchmark** | ARC | 0.534 | 0.425 | 0.522 | 0.373 | 0.583 | 0.460 |
| | HellaSwag | 0.284 | 0.206 | 0.247 | 0.177 | 0.373 | 0.265 |
| | MMLU | 0.579 | 0.437 | 0.567 | 0.408 | 0.651 | 0.507 |
| | TruthfulQA | 0.652 | 0.484 | 0.650 | 0.538 | 0.741 | 0.590 |
| | Winogrande | 0.439 | 0.307 | 0.383 | 0.278 | 0.480 | 0.336 |
| | GSM8K | 0.202 | 0.159 | 0.256 | 0.165 | 0.222 | 0.147 |
| | AVG | 0.575 | 0.429 | 0.596 | 0.425 | 0.619 | 0.476 |
| **Evaluation Method Comparison** | | | | | | | |
| **EHRNoteQA Open-Ended** | GPT-4 Eval | **0.770** | **0.609** | **0.805** | **0.617** | **0.801** | **0.657** |
| | BLEU | 0.155 | 0.112 | 0.037 | 0.059 | 0.014 | -0.006 |
| | ROUGE-L | 0.500 | 0.324 | 0.398 | 0.283 | 0.356 | 0.241 |
| | Exact Match | 0.422 | 0.288 | 0.336 | 0.236 | 0.266 | 0.194 |
| | SentenceBERT | 0.710 | 0.524 | 0.726 | 0.555 | 0.652 | 0.453 |
| | ClinicalBERT | 0.536 | 0.382 | 0.552 | 0.389 | 0.394 | 0.288 |
| **EHRNoteQA Multi-Choice** | GPT-4 Eval | **0.766** | **0.661** | **0.732** | **0.574** | **0.812** | **0.661** |
| | Probability(index) | 0.622 | 0.472 | 0.596 | 0.444 | 0.676 | 0.519 |
| | Probability(value) | 0.514 | 0.437 | 0.523 | 0.456 | 0.549 | 0.437 |

key question arises: Would a model that scores highly on EHRNoteQA also receive high scores from clinicians' evaluation? In other words, can EHRNoteQA serve as a reliable proxy for actual expert evaluation? Furthermore, how would other benchmarks, such as existing discharge summary QA and clinical knowledge benchmarks, compare to EHRNoteQA in representing real-world assessment? To address these questions, we conduct an additional experiment with 19 LLMs[5], measuring the correlation between LLM performance manually evaluated by clinicians, and LLM performance automatically evaluated on diverse benchmarks including EHRNoteQA.

For manually evaluating LLM performance, three clinicians assessed the LLM responses to discharge summary-based questions, resulting in individual scores for each of the 19 models. For the questions, we used DiSCQ [30], a collection of questions asked by medical experts based on the MIMIC-III discharge summaries [25]. The medical experts involved in DiSCQ are different from those who participated in the creation of EHRNoteQA to avoid potential bias. From a total of 1,089 DiSCQ questions, 300 were randomly selected, with each clinician assigned to 100 distinct questions. Each clinician evaluated the responses of the 19 LLMs regarding the 100 DiSCQ questions (a total of 1,900 responses), marking responses as either correct or incorrect based on the corresponding discharge summaries. Through this process, we obtained each score of the 19 LLMs from each of the three clinicians, referred to as *clinician-evaluated LLM scores* in the paper. The DiSCQ questions assigned to each clinician and their evaluated scores for the models can be found in the Appendix F.

We then measure the correlation between these *clinician-evaluated LLM scores* against the automatically evaluated LLM scores on EHRNoteQA Level 1 data and several other benchmarks. These

---

[5]Among the 27 models in Table 4, three GPT models were excluded because their general benchmark scores are unavailable on the HuggingFace LLM leaderboard. Additionally, three models (*Llama3-70b, Mixtral-8x7b, Gemma-7b*) were released after our experiment ended and thus are not included. Lastly, two models (*Asclepius-7b, Asclepius-13b*) were excluded due to their inability to perform multi-choice QA.

benchmarks include data from previous studies on discharge summary QA (emrQA [44] and Yue et al. [65]), as well as clinical domain knowledge benchmarks commonly used in LLM research (MedQA [23], PubMedQA [24], MMLU* [19] [6], MedMCQA [43]), and general domain benchmarks from the HuggingFace open-LLM Leaderboard [4] (ARC [10], HellaSwag [66], MMLU [19], TruthfulQA [35] ,Winogrande [49], GSM8K [11]) for comprehensive comparison. Following our EHRNoteQA evaluation method, we use the same automated evaluation method using GPT-4 for the discharge summary QA datasets (emrQA and Yue et al. [65]) and clinical benchmarks (MedQA, PubMedQA, MMLU*, MedMCQA), while general benchmarks (ARC, HellaSwag) scores are obtained directly from the Huggingface open-LLM leaderboard [4]. Specific scores are detailed in Appendix F.3.

The correlations are presented in Table 5. Each column shows the Spearman($\rho$) and Kendall($\tau$) correlation coefficients between the *clinician-evaluated LLM scores* and the benchmark scores. First, when measuring the correlation between the clinician-evaluated LLM scores of clinician A, B, and C, we find high intra-clinician correlations. Such results provide preliminary credibility of clinician evaluations, as each clinician evaluated a different subset of data. Notably, EHRNoteQA demonstrates the highest correlation in both multi-choice and free-text formats for all clinicians, consistently outperforming all other benchmarks. These results highlight the efficacy of EHRNoteQA in evaluating LLM performance within the specified clinical context, compared to other benchmarks.

Lastly, we reconfirm the validity of our GPT-4-based evaluation method for EHRNoteQA by comparing the correlations between *clinician-evaluated LLM scores* and EHRNoteQA scores obtained through other evaluation methods. For open-ended questions, when evaluating with traditional methods such as BLEU [45], ROUGE-L [34], Exact Match, and cosine similarity (using Sentence-BERT [48] and ClinicalBERT [2]), the results show lower correlation compared to our GPT-4 based evaluation scores. For multi-choice questions, many studies employ probability-based scoring. Following this approach, we test two types of probability-based scoring methods on EHRNoteQA using *LM-Evaluation Harness* [17]: one measuring the probability of the answer index (e.g., A) and another measuring the probability of the correct answer choice text. However, both results show lower correlation compared to our GPT-4 based evaluation scores.

## 5 Discussion

**Limitations.** The model performance on EHRNoteQA may not directly align with the model performance when evaluating on EHR discharge summaries outside of the MIMIC dataset. EHRNoteQA is built upon the MIMIC-IV dataset, and when computing its correlation with real-world clinical practice, we used questions derived from the MIMIC-III dataset. This experiment setting was inevitable because MIMIC is currently the only publicly available EHR dataset accessible for research purposes. However, it is important to note that if clinical institutions aim to apply LLMs to their own EHR systems, EHRNoteQA can serve as an initial benchmark for their model selection.

**Future Direction.** The current version of EHRNoteQA is tailored to align with the context length of currently available LLMs, categorizing the data into Level 1 (4k tokens) and Level 2 (8k tokens). As models capable of handling longer context lengths are developed and released, we plan to extend EHRNoteQA to include datasets with more admissions and longer discharge summaries. Additionally, while EHRNoteQA focuses on discharge summaries due to their frequent use in clinical practice, we recognize the importance of expanding our dataset to include other types of notes essential in healthcare settings, such as radiology notes and physician notes.

## Acknowledgments and Disclosure of Funding

This work was supported by the KAIST-NAVER Hyper-Creative AI Center, the Institute of Information & Communications Technology Planning & Evaluation (IITP) grant (No.RS-2019-II190075), National Research Foundation of Korea (NRF) grant (NRF-2020H1D3A2A03100945), and Korea Medical Device Development Fund grant (Project Number: 1711138160, KMDF_PR_20200901_0097), funded by the Korea government (MSIT, MOTIE, MOHW, MFDS).

---

[6]Selected 6 clinical topics from the 57 subjects in MMLU: *Anatomy, Clinical Knowledge, College Biology, College Medicine, Medical Genetics, Professional Medicine*.

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

## Supplementary Contents

# A  Datasheet for Datasets

## A.1  Motivation

- **For what purpose was the dataset created?**

  We created EHRNoteQA to evaluate Large Language Models (LLMs) in real-world clinical scenarios for answering clinicians' questions regarding patient discharge summaries.

- **Who created the dataset (e.g., which team, research group) and on behalf of which entity (e.g., company, institution, organization)?**

  The authors of the paper.

- **Who funded the creation of the dataset? If there is an associated grant, please provide the name of the grantor and the grant name and number.**

  This work was supported by the KAIST-NAVER Hyper-Creative AI Center, and the National Research Foundation of Korea (NRF) grant (NRF-2020H1D3A2A03100945) funded by the Korea government (MSIT)

## A.2  Composition

- **What do the instances that comprise the dataset represent (e.g., documents, photos, people, countries)?**

  EHRNoteQA contains natural language questions along with their corresponding answers in both open-ended and multiple-choice formats (text).

- **How many instances are there in total (of each type, if appropriate)?**

  EHRNoteQA consists of 962 question-answer pairs.

- **Does the dataset contain all possible instances or is it a sample (not necessarily random) of instances from a larger set?**

  The dataset contains all instances of the data.

- **What data does each instance consist of?**

  EHRNoteQA consists of a (Question, Answer) pair for each instance. "Answer" includes the correct answer and four incorrect answer choices for both open-ended and multi-choice evaluations.

- **Is there a label or target associated with each instance?**

  The answer (label) is provided for each instance.

- **Is any information missing from individual instances? If so, please provide a description, explaining why this information is missing (e.g., because it was unavailable). This does not include intentionally removed information, but might include, e.g., redacted text.**

  No.

- **Are relationships between individual instances made explicit (e.g., users' movie ratings, social network links)?**

  N/A.

- **Are there recommended data splits (e.g., training, development/validation, testing)?**

  No, EHRNoteQA is an evaluation benchmark that solely consists of test data.

- **Are there any errors, sources of noise, or redundancies in the dataset?**

  Each question-answer pair is manually reviewed and modified by clinicians. Despite our efforts, some questions or answers may contain minor grammatical errors, but these are not critical.

- **Is the dataset self-contained, or does it link to or otherwise rely on external resources (e.g., websites, tweets, other datasets)?**

  EHRNoteQA links to the MIMIC-IV EHR database[7], which is accessible via PhysioNet[8].

---

[7] https://physionet.org/content/mimic-iv-note/2.2/
[8] https://physionet.org

- **Does the dataset contain data that might be considered confidential (e.g., data that is protected by legal privilege or by doctor-patient confidentiality, data that includes the content of individuals' non-public communications)?**

  No.

- **Does the dataset contain data that, if viewed directly, might be offensive, insulting, threatening, or might otherwise cause anxiety?**

  No.

- **Does the dataset relate to people?**

  Yes.

- **Does the dataset identify any subpopulations (e.g., by age, gender)?**

  EHRNoteQA is based on patients from the MIMIC-IV EHR database, which includes 145,915 patients with discharge summaries from Beth Israel Deaconess Medical Center (BIDMC). These anonymized patient records span from 2008 to 2019.

- **Does the dataset contain data that might be considered sensitive in any way (e.g., data that reveals race or ethnic origins, sexual orientations, religious beliefs, political opinions or union memberships, or locations; financial or health data; biometric or genetic data; forms of government identification, such as social security numbers; criminal history)?**

  No. The source dataset is already de-identified.

## A.3 Collection process

- **How was the data associated with each instance acquired?**

  For the initial draft of the dataset, we used GPT-4 to generate each QA pair on Azure's HIPAA-compliant platform, adhering to MIMIC-IV data usage regulations. Subsequently, three clinicians thoroughly reviewed and modified each question-answer pair to ensure clinical relevance.

- **What mechanisms or procedures were used to collect the data (e.g., hardware apparatuses or sensors, manual human curation, software programs, software APIs)?**

  We used GPT-4 on Azure's HIPAA-compliant platform to generate the initial draft of the dataset. Furthermore, three clinicians were involved to manually review and modify the dataset for clinical relevance.

- **If the dataset is a sample from a larger set, what was the sampling strategy (e.g., deterministic, probabilistic with specific sampling probabilities)?**

  From the filtered patients (see Section 3.1), we randomly sampled a total of 962 patients.

- **Who was involved in the data collection process (e.g., students, crowd workers, contractors) and how were they compensated (e.g., how much were crowd workers paid)?**

  The data was constructed exclusively by the authors of the study. No crowd workers or external contractors were involved.

- **Over what timeframe was the data collected?**

  EHRNoteQA was constructed between 2023 and 2024. It was built using the MIMIC-IV EHR database, which consists of patient records from 2008 to 2019.

- **Were any ethical review processes conducted (e.g., by an institutional review board)?**

  No.

- **Does the dataset relate to people?**

  Yes.

- **Did you collect the data from the individuals in question directly, or obtain it via third parties or other sources (e.g., websites)?**

  N/A.

- **Were the individuals in question notified about the data collection?**

  N/A.

- **Did the individuals in question consent to the collection and use of their data?**
  N/A.

- **If consent was obtained, were the consenting individuals provided with a mechanism to revoke their consent in the future or for certain uses?**
  N/A.

- **Has an analysis of the potential impact of the dataset and its use on data subjects (e.g., a data protection impact analysis) been conducted?**
  The dataset is anonymized and does not contain any personal patient information.

### A.4 Preprocessing/cleaning/labeling

- **Was any preprocessing/cleaning/labeling of the data done (e.g., discretization or bucketing, tokenization, part-of-speech tagging, SIFT feature extraction, removal of instances, processing of missing values)?**
  The data was manually reviewed and modified by clinicians to ensure clinical relevance. This process includes deleting questions that are not clinically important and modifying the questions, answers, and incorrect answer choices.

- **Was the "raw" data saved in addition to the preprocess/cleaned/labeled data (e.g., to support unanticipated future uses)?**
  N/A.

- **Is the software that was used to preprocess/clean/label the data available?**
  N/A.

### A.5 Uses

- **Has the dataset been used for any tasks already?**
  No.

- **Is there a repository that links to any or all papers or systems that use the dataset?**
  No.

- **What (other) tasks could the dataset be used for?**
  The dataset is designed to promote research in question answering on patient EHRs.

- **Is there anything about the composition of the dataset or the way it was collected and preprocessed/cleaned/labeled that might impact future uses?**
  N/A.

- **Are there tasks for which the dataset should not be used?**
  N/A.

### A.6 Distribution

- **Will the dataset be distributed to third parties outside of the entity (e.g., company, institution, organization) on behalf of which the dataset was created?**
  No.

- **How will the dataset be distributed?**
  The dataset will be released via GitHub upon publication.

- **Will the dataset be distributed under a copyright or other intellectual property (IP) license, and/or under applicable terms of use (ToU)?**
  The dataset is released under MIT license.

- **Have any third parties imposed IP-based or other restrictions on the data associated with the instances?**
  No.

- **Do any export controls or other regulatory restrictions apply to the dataset or to individual instances?**
  No.

### A.7 Maintenance

- **Who will be supporting/hosting/maintaining the dataset?**
  The authors of the paper.

- **How can the owner/curator/manager of the dataset be contacted(e.g., email address)?**
  Contact the first authors (`sean0042@kaist.ac.kr` & `jiyoun.kim@kaist.ac.kr`).

- **Is there an erratum?**
  No.

- **Will the dataset be updated (e.g., to correct labeling erros, add new instances, delete instances)?**
  If any corrections are required, we plan to upload an updated version of the dataset along with specific explanations of the updates. Additionally, we intend to consistently update our dataset as mentioned in the future directions in Section 5.

- **If the dataset relates to people, are there applicable limits on the retention of the data associated with the instances (e.g., were the individuals in question told that their data would be retained for a fixed period of time and then deleted)?**
  N/A.

- **Will older versions of the dataset continue to be supported/hosted/maintained?**
  We plan to maintain only the most recent version of the dataset. However, if there are significant updates, we will preserve the previous versions.

- **If others want to extend/augment/build on/contribute to the dataset, is there a mechanism for them to do so?**
  Contact the first authors of the paper.

# B   Preliminary

## B.1   Data Resources

The following clinical datasets were utilized in compliance with the PhysioNet license, ensuring adherence to ethical guidelines for the use of medical data.

- MIMIC-IV Discharge Summary [26]: Available at https://physionet.org/content/mimic-iv-note/2.2/
- MIMIC-III Discharge Summary [25]: Available at https://physionet.org/content/mimiciii/1.4/
- DiSCQ [30] : Available at https://physionet.org/content/discq/1.0/
- Yue et al. [64] : Available at https://physionet.org/content/mimic-iii-question-answer/1.0.0/

The following clinical data was sourced under the license from the Department of Biomedical Informatics (DBMI) at the Blavatnik Institute of Harvard Medical School.

- emrQA [44] : Available at https://portal.dbmi.hms.harvard.edu/projects/n2c2-nlp/

Open-source datasets from Huggingface [61].

- MedQA [23] : Available at https://huggingface.co/datasets/bigbio/med_qa
- MedMCQA [24] : Available at https://huggingface.co/datasets/openlifescienceai/medmcqa
- PubmedQA [43] : Available at https://huggingface.co/datasets/bigbio/pubmed_qa
- MMLU [19] : Available at https://huggingface.co/datasets/cais/mmlu

The performance scores of various models on additional benchmarks—such as ARC [10], Hellaswag [66], MMLU [19], TruthfulQA [35], Winogrande [49], GSM8k [11], and the overall average score—were sourced from the Open-Source LLM Leaderboard [4].

- Open-LLM-Leaderboard [4] : Available at https://huggingface.co/spaces/open-llm-leaderboard/open_llm_leaderboard

## B.2   Model Resources

The models used in this paper are divided into closed-source and open-source models. The closed-source models, GPT series (GPT-4, GPT-4-Turbo, GPT-3.5-Turbo), are all run on Azure's HIPAA-compliant platform[9], addressing the potential privacy leakage issues that could arise from using clinical data. The specific model IDs for the GPT series in Table 4 are as follows:

- GPT4 : `GPT4 (0613)`
- GPT4-Turbo : `GPT4-turbo-preview (1106)`
- GPT3.5-Turbo : `GPT3.5-turbo-16k (0613)`

The open-source models listed in Table 4 are all downloaded from Huggingface [61] and run locally through direct inference. During inference, the 7B and 13B models were run on 1-2 RTX A6000 GPUs, the 30B and 8x7B models were run on 2-4 RTX A6000 GPUs, and the 70B models were run on 4-8 RTX A6000 GPUs, with the exact number of required GPUs depending on the input length (e.g., Level 1, Level 2 data of EHRNoteQA). Table 6 provides the Huggingface paths for each model.

---

[9]https://learn.microsoft.com/en-us/azure/compliance/offerings/offering-hipaa-us

Table 6: HuggingFace paths for each open-source model

| Size | Model | Huggingface Path | Reference |
|---|---|---|---|
| 70B | Llama3-70b-Instruct | meta-llama/Meta-Llama-3-70B-Instruct | [1] |
| | Llama2-70b-chat | meta-llama/Llama-2-7b-chat-hf | [56] |
| | qCammel-70 | augtoma/qCammel-13 | [55] |
| | Camel-Platypus2-70b | garage-bAInd/Camel-Platypus2-70B | [28] |
| | Platypus2-70b-Instruct | garage-bAInd/Platypus2-70B-instruct | [28] |
| 8x7B | Mixtral-8x7b-Instruct | mistralai/Mixtral-8x7B-Instruct-v0.1 | [22] |
| 30B | MPT-30b-Instruct | mosaicml/mpt-30b-instruct | [39] |
| 13B | Llama2-13b-chat | meta-llama/Llama-2-13b-chat-hf | [56] |
| | Vicuna-13b | lmsys/vicuna-13b-v1.5 | [9] |
| | WizardLM-13b | WizardLMTeam/WizardLM-13B-V1.0 | [62] |
| | qCammel-13 | augtoma/qCammel-13 | [55] |
| | OpenOrca-Platypus2-13b | Open-Orca/OpenOrca-Platypus2-13B | [29] |
| | Camel-Platypus2-13b | garage-bAInd/Camel-Platypus2-13B | [28] |
| | Synthia-13b | migtissera/Synthia-13B-v1.2 | [54] |
| | Asclepius-13b | starmpcc/Asclepius-Llama2-13B | [27] |
| 7B | Gemma-7b-it | google/gemma-7b-it | [52] |
| | MPT-7b-8k-instruct | mosaicml/mpt-7b-instruct | [39] |
| | Mistral-7b-Instruct | mistralai/Mistral-7B-Instruct-v0.2 | [21] |
| | Dolphin-2.0-mistral-7b | cognitivecomputations/dolphin-2.0-mistral-7b | [12] |
| | Mistral-7b-OpenOrca | Open-Orca/Mistral-7B-OpenOrca | [31] |
| | SynthIA-7b | migtissera/SynthIA-7B-v1.3 | [53] |
| | Llama2-7b-chat | meta-llama/Llama-2-7b-chat-hf | [56] |
| | Vicuna-7b | lmsys/vicuna-7b-v1.5 | [9] |
| | Asclepius-7b | starmpcc/Asclepius-Llama2-7B | [27] |

# C EHRNoteQA Construction

## C.1 Preprocessing Discharge Summaries

To create the EHRNoteQA dataset, we use MIMIC-IV discharge summaries [26]. As mentioned in Section 3.1, the length of these summaries is lengthy and exceeds the context length limit of current LLMs. Thus, we first reduce the excessive white space using a regular expression, which decreases the average total note length by 10%. Listing C.1 provides the Python code used to perform this task.

```python
import re

def transform_string(s):
    s = re.sub(r'(\n\s*|\s*\n)', '\n', s)
    s = re.sub(r'\s{2,}', ' ', s)
    s = s.strip()
    return s
```

Listing 1: Python code for reducing excessive white spaces

Then, we add the metadata at the beginning of the discharge summary: patient ID, admission ID, and chart time. Without this information, it would be impossible to distinguish the sequence of multiple discharge summaries for a single patient when feeding them into the model. This metadata is obtained from the `subject_id`, `hadm_id`, and `charttime` columns provided in the discharge summary CSV file from MIMIC-IV. An example of the whole preprocessing step is described in Figure 2.

Figure 2: Preprocessing MIMIC-IV discharge summaries. First, we remove excessive white spaces which lead to average 10% token reduction (middle). Then, we add meta information (Patient ID, Admission ID, and Chartdate) in front of each note (end).

## C.2 GPT-4 Usage in Data Generation

The initial QA data was created using GPT-4 [42]. Specifically, we input discharge summaries from MIMIC-IV into GPT-4 to generate each QA pair. The QA generation process involves a two-step approach: Question Generation and Answer Generation. In Question Generation phase, a clinically relevant question that clinicians may ask is generated, regarding the given patient discharge summaries. Subsequently, in Answer Generation phase, the corresponding answer along with four incorrect answer choices is generated. To ensure the quality of the generated data, we collaborated closely with clinicians during the prompt tuning phase. Refer to Figures 3 and 4 for the specific prompts.

GPT-4 was accessed through Azure's HIPAA-compliant platform, adhering to the usage guidelines of the MIMIC-IV database in Physionet. We utilized the GPT-4 (0613) model version with default hyperparameters, except for the temperature, which was set to 1. The cost for generating a total of 1,000 data samples was approximately $3,000.

## C.3 Clinician Review & Modification

As mentioned in Section 3.3, the initial 1,000 QA pairs generated by GPT-4 underwent a thorough review and modification process by three clinicians. Each clinician was assigned 333 / 333 / 334 QA pairs, respectively. The clinicians were provided with each QA pair along with the corresponding patient's discharge summaries. The clinicians were instructed to process the data as follows:

1. **Question Elimination**: First, assess whether the question generated by GPT-4 is clinically meaningful for the patient. Additionally, evaluate if the question is not one that a physician would typically ask. If either of these criteria are not met, mark the question with an "X" for elimination.

2. **Question Modification**: For questions that meet the initial criteria, proceed to revise them if necessary. Specifically, if a question is ambiguously phrased or requests excessive information, rephrase it for clarity and conciseness.

3. **Answer Modification**: For all questions, whether modified or not during the Question Modification phase, review and update the corresponding answer. Ensure that the answer is complete, covering all necessary information.

4. **Incorrect Answer Choice Modification**: Modify the four incorrect answer choices to make them more challenging. Instead of providing choices that are immediately recognizable as incorrect without referring to the discharge summaries, create plausible distractors based on the discharge summaries that are not the correct answers to the question.

Streamlit[10] and Google Sheets were used to facilitate modifications by the three clinicians. Streamlit was used to display the questions, answer choices, and discharge summaries, while Google Sheets was used to receive the clinicians' revisions.

Every ten days, each clinician was assigned to revise 50 QA pairs, and the entire revision process took two months. Figures 5 and 6 show screenshots of the Streamlit interface and Google Sheets provided to the clinicians, respectively. Additional examples of QA revisions can be seen in Table 7 and 8.

---

[10]https://streamlit.io/

**Question Generation Prompt**

Situation :
When a patient is admitted to the hospital, important clinical records are summarized in a 'discharge summary'. On the patient's subsequent visit, the previous 'discharge summaries' serve as essential reference for the doctor's clinical decision making.

Objective :
Please formulate one question that a doctor might actually ask based on the provided 'discharge summaries'. The questions should have clear answer, and these answer should be found within the provided 'discharge summaries'.

Note:
- The 'discharge summary' is provided between [note 1 start] and [note 1 end]. If there are multiple notes, they are labeled as [note 1 start], [note 2 start], etc.
- The 'discharge summaries' are provided in chronological order. This means note1 is a record from before note2. At the beginning of each note, there is an admission ID and the date it was written, so please refer to that.
- Please refrain from formulating questions that can be answered without referring to a note.
- Do not create question that is too easy to answer. To answer your question, someone should have the clinical expertise equivalent to a doctor and must fully understand all provided discharge summaries.
- Your answer should also contain short rationale behind the answer.
- When explaining the answer and its rationale, utilize the chart date of the note. In other words, instead of saying the first note or the second note, phrase it as 'as per the note charted on [specific date], ...'.
- Arrange your output in the following format:
  - Question : [Your Question]
  - Answer : [Your Answer]
  - Reason : [Explanation for your answer]

[note 1 start]
{{discharge_summary_1}}
[note 1 end]

[note 2 start]
{{discharge_summary_2}}
[note 2 end]

[note 3 start]
{{discharge_summary_3}}
[note 3 end]

Figure 3: Prompt template for question generation of EHRNoteQA (1st step)

**Answer Generation Prompt**

Objective :
Please generate a multiple-choice question answering data with five possible answers (A-E) based on the doctor's question derived from the provided 'discharge summary'. Ensure that one answer is correct, and the remaining four are incorrect answers.

Note:

- Use the provided doctor's question as the basis for the multiple-choice question without modification.

- The 'discharge summary' is enclosed within [note 1 start] and [note 1 end], with additional notes being similarly labeled.

- All distractors (incorrect answer choices) should contain contents that appear in the provided discharge summary but should be wrong answer to the question.

- After choosing all five choices, paraphrase them so that all choices are consistent with the format and length. Ensure that longest length answer choice is not an answer

- The correct answer should be clearly indicated, and the rationale should explain why this is the answer and why the other options are not correct but are good distractors.

- Arrange your output in the following format:
  - Question: [The doctor's question]
  - Answer Choices:

    – A: [First option]
    – B: [Second option]
    – C: [Third option]
    – D: [Fourth option]
    – E: [Fifth option]

  - Correct Answer: [The letter of the correct choice]
  - Reason: [Explanation behind your answer and why each other options are incorrect but can be good distractors]

[note 1 start]
{{discharge_summary_1}}
[note 1 end]

[note 2 start]
{{discharge_summary_2}}
[note 2 end]

[note 3 start]
{{discharge_summary_3}}
[note 3 end]

Figure 4: Prompt template for answer generation of EHRNoteQA (2nd step)

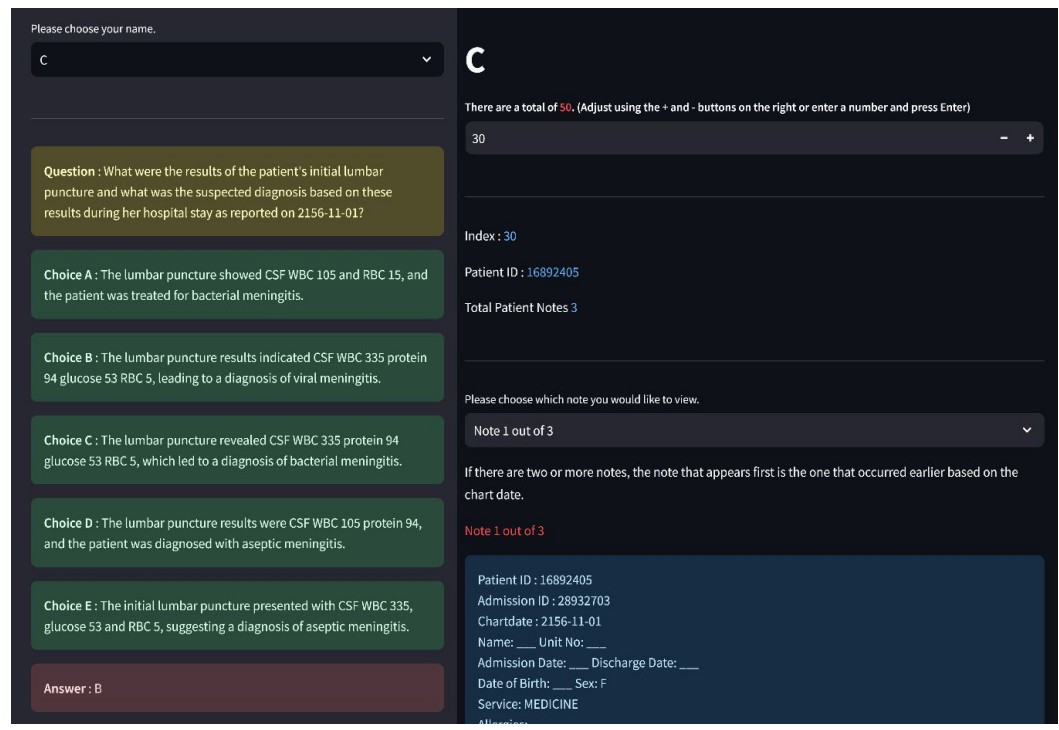

Figure 5: Screenshot of Streamlit provided to clinicians for EHRNoteQA review & modification.

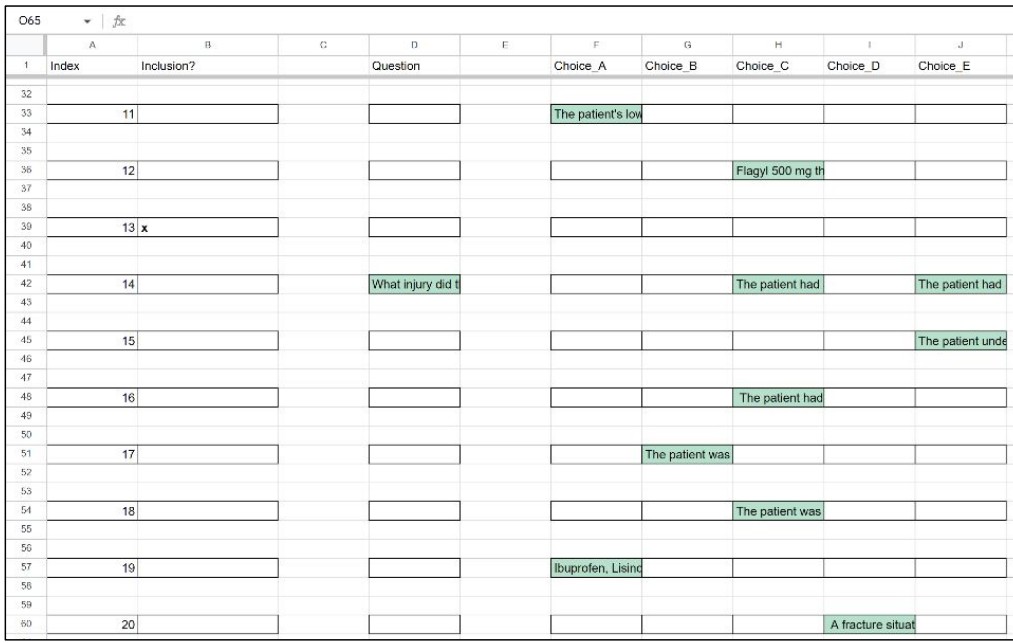

Figure 6: Screenshot of Google Sheet provided to clinicians for EHRNoteQA review & modification.

Table 7: Examples of before and after clinician review & modifications for EHRNoteQA

| Before |
|---|
| **Question**: What medical procedures were performed on the patient's right vs. left leg, and when? |
| **Correct Answer**: The patient underwent internal fixation of right tibial fracture and internal fixation of left fibula fracture, unspecified date. |
| **Incorrect Answer Choices**: |
| - The patient underwent knee surgery on her left leg, unspecified date. |
| - The patient had surgical removal of left ankle hardware, unspecified date. |
| - The patient underwent open reduction of ankle dislocation on her right leg, unspecified date. |
| - The patient underwent tibial fracture stabilization in her left leg, unspecified date. |
| **After** |
| **Question**: What surgical procedures were performed on the patient's right vs. left leg, and when? |
| **Correct Answer**: The patient underwent internal fixation of both tibial fracture and internal fixation of left fibula fracture, unspecified date. |
| **Incorrect Answer Choices**: |
| - The patient underwent knee surgery on her left leg, unspecified date. |
| - The patient had surgical removal of left ankle hardware, unspecified date. |
| - The patient underwent open reduction of ankle dislocation on her right leg, unspecified date. |
| - The patient underwent tibial fracture stabilization in her left leg, unspecified date. |
| **Before** |
| **Question**: What subcapsular injury did the patient sustain from his mechanical fall and how was it managed during their hospital stay? |
| **Correct Answer**: This patient had a small subcapsular liver hematoma, managed by following chest tube output and performing serial abdominal exams. |
| **Incorrect Answer Choices**: |
| - The patient broke his right rib and it was managed by bandaging the area tightly. |
| - The patient suffered from a subcapsular kidney hematoma which was managed by constant kidney flushing. |
| - The patient had a subcapsular spleen injury managed with a series of prescribed antibiotics. |
| - The patient had a subcapsular heart injury, managed through a series of minor surgeries. |
| **After** |
| **Question**: What injury did the patient sustain from his mechanical fall and how was it managed during their hospital stay? |
| **Correct Answer**: This patient had a small subcapsular liver hematoma, managed by following chest tube output and performing serial abdominal exams. |
| **Incorrect Answer Choices**: |
| - The patient underwent knee surgery on her left leg, unspecified date. |
| - The patient had surgical removal of left ankle hardware, unspecified date. |
| - The patient had a subcapsular spleen injury managed with a chest tube insertion. |
| - The patient had a subcapsular liver injury, managed through a series of minor surgeries. |
| **Before** |
| **Question**: What was the patient's diagnosis upon discharge after his second admission? |
| **Correct Answer**: Seizure disorder and Polysubstance abuse |
| **Incorrect Answer Choices**: |
| - Fibromyalgia |
| - Corrosive substance abuse |
| - Epilepsy |
| - Coronary artery disease |
| **After** |
| **Question**: What was the patient's diagnosis upon discharge after his second admission? |
| **Correct Answer**: Constipation |
| **Incorrect Answer Choices**: |
| - Fibromyalgia |
| - Seizure |
| - Epilepsy |
| - Coronary artery disease |

Table 8: Examples of before and after clinician review & modifications for EHRNoteQA (continued)

| Before |
|---|

**Question**: What were the post-operative complications experienced by the patient after the first surgery and how were they managed?

**Correct Answer**: The patient had an ileus with no evidence of obstruction. This was managed by making the patient follow a nil per mouth (NPO) regime, providing IV hydration, and an aggressive bowel treatment until the patient had a bowel movement and flatus.

**Incorrect Answer Choices**:

- The patient developed severe pain which was managed through progressive introduction of oral pain medications.

- The patient developed an obstruction that was managed through an aggressive bowel regimen.

- The patient experienced adverse reactions to medication which were managed by altering the prescription.

- The patient had severe bleeding that was controlled by medication and medical intervention.

| After |
|---|

**Question**: What were the post-operative complications experienced by the patient after the first surgery and how were they managed?

**Correct Answer**: The patient had an ileus with no evidence of obstruction. This was managed by suppertive care until the patient had a bowel movement and flatus.

**Incorrect Answer Choices**:

- The patient developed ileus which was managed through IV and PO meds.

- The patient developed an obstructive ileus that was managed through an aggressive bowel regimen.

- The patient experienced adverse reactions to medication which were managed by altering the prescription.

- The patient had severe bleeding that was controlled by medication and medical intervention.

| Before |
|---|

**Question**: What changes were made to the patient's prescription to manage his orthostatic hypotension and hypertension between his visits in 2161 and 2168?

**Correct Answer**: Hydralazine was discontinued and labetalol and amlodipine doses were not changed.

**Incorrect Answer Choices**:

- The medication Irbesartan was increased from 150 mg daily to 37.5 mg daily.

- Folic Acid dosage was increased to manage orthostatic hypotension and hypertension.

- Calcium Acetate was replaced by a new medication to manage blood pressure.

- The medication Labetalol was discontinued.

| After |
|---|

**Question**: What changes were made to the patient's prescription to manage his orthostatic hypotension during his visits in 2161?

**Correct Answer**: Hydralazine was discontinued and labetalol and amlodipine doses were not changed.

**Incorrect Answer Choices**:

- The medication Irbesartan was increased from 150 mg daily to 37.5 mg daily.

- Folic Acid dosage was increased to manage orthostatic hypotension and hypertension.

- All hypertensive medications were discontinued.

- The medication Labetalol was discontinued.

| Before |
|---|

**Question**: What was the medical intervention made to treat the patient's adnexal mass issue on her first visit?

**Correct Answer**: She underwent a total abdominal hysterectomy and a bilateral salpingo-oophorectomy.

**Incorrect Answer Choices**:

- A laparoscopic tubal ligation was done.

- The patient was put on a regimen of Nabumetone, Fluoxetine, Clonazepam, Trazodone, and Baclofen.

- A partial thyroidectomy was performed.

- The patient was given discharge medications like Trazodone and Clonazepam.

| After |
|---|

**Question**: What was treatment for patient's adnexal mass issue on her first visit?

**Correct Answer**: She underwent a total abdominal hysterectomy and a bilateral salpingo-oophorectomy.

**Incorrect Answer Choices**:

- A laparoscopic tubal ligation was done.

- The patient was put on a regimen of Nabumetone, Fluoxetine, Clonazepam, Trazodone, and Baclofen.

- A partial thyroidectomy was performed.

- She underwent a total abdominal hysterectomy and a right salpingo-oophorectomy.

## D EHRNoteQA Categorization

For the final 962 EHRNoteQA data, the authors manually reviewed and categorized all questions based on the criteria below. These categories and criteria were determined in consultation with clinicians.

- **Treatment** : How a patient's problem is managed.
- **Assessment** : A patient's status change or his/her condition.
- **Problem** : Problems that a patient had experienced.
- **Etiology** : Causes of a patient's symptom and problem.
- **Sign/Symptom** : Symptoms of a patient.
- **Vitals** : Body temperature, pulse rate, respiration rate, or blood pressure of a patient.
- **Test Results** : Diagnostic test results used to monitor disease progression.
- **History** : A patient's personal medical history or family medical history.
- **Instruction** : Instruction and advice given to a patient by the hospital after surgery or procedure.
- **Plan** : Future treatment plans considering a patient's condition.

According to the criteria mentioned above, the proportions of the 962 categorized questions can be seen in Figure 7. Additionally, examples of questions for each category can be found in Table 9. Note that the proportions do not sum to 100% because a single question can be categorized into multiple categories if it contains more than one topic *(e.g., Etiology and Treatment: "What were the primary causes of the patient's shortness of breath and how were these managed during the hospital stay?")*.

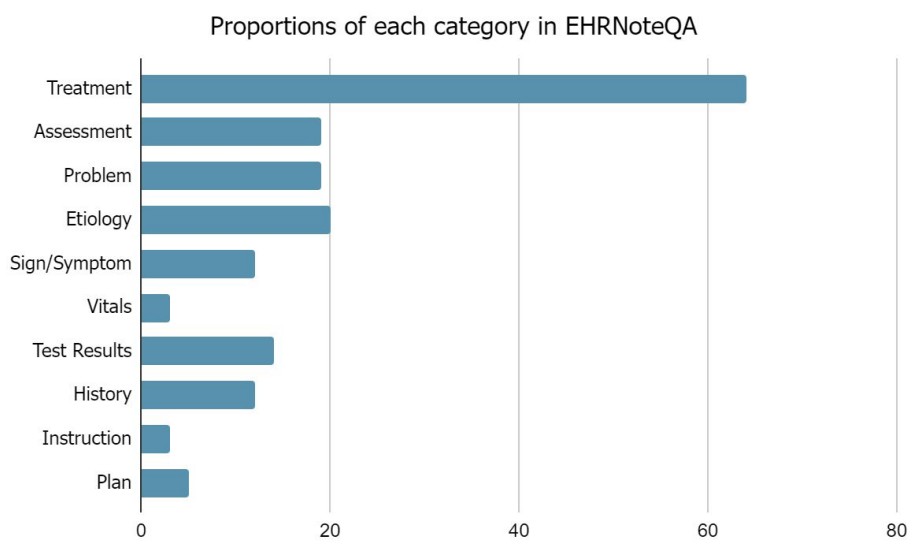

Figure 7: Barplot showing the proportions(%) of each category in the EHRNoteQA.

Table 9: Examples of EHRNoteQA questions for each category.

| Index | Category | Example |
|---|---|---|
| 1 | Treatment | Based on the patient's discharge information, what invasive procedures have been performed to address the patient's cervical stenosis and related symptoms and what medication adjustments were made in response to those surgical outcomes? |
| 2 | Treatment | Which coronary artery of the patient was stented? |
| 3 | Treatment | Which medication was prescribed to the patient with a frequency of three times a day at the time of discharge? |
| 4 | Assessment | How were his mental status changes progress over time between his hospital stays? |
| 5 | Assessment | Did the patient's abdominal pain and elevated Liver Function Tests (LFTs) resolve after the ERCP procedure and what other ailments were observed during her hospital stay? |
| 6 | Assessment | Regarding the patient's current condition, what kind of surgical procedure did she undergo and what are the characteristics of the lesion identified in her pancreas? |
| 7 | Problem | Did the patient experience any complications following either their Laparoscopic subtotal colectomy and Laparoscopic total abdominal hysterectomy procedures? |
| 8 | Problem | Given the patient's history with High-grade Dysplasia, did he experience any complications post his laparoscopic right colectomy? |
| 9 | Problem | Has the patient experienced any allergic reactions to antibiotics and if so, which specific antibiotics are listed in his clinical history? |
| 10 | Etiology | Was there any evidence, from the discharge summary dated 2132-09-22, to suggest that the patient's diplopia might have been caused by a vertebral artery dissection? |
| 11 | Etiology | How does the patient's liver hemangioma status relate to her later incident of left breast hematoma? What was the cause behind left breast hematoma? |
| 12 | Etiology | Why did the patient's creatinine level rise significantly upon admission, and how was it addressed during his hospital stay? |
| 13 | Sign/Symptom | Based on the patient's past medical history and discharge summary, what were the symptoms the patient presented at the hospital and what was the final cause of the patient's death? |
| 14 | Sign/Symptom | Was there any change in patient's symptoms related to pain and bowel movements from first admission on 2123-05-10 to the second admission on 2123-07-27 and finally to the last admission on 2123-09-24? |
| 15 | Sign/Symptom | Has the patient experienced chest pain, shortness of breath or lightheadedness related to exertion or exertion related activities between her hysterectomy surgery and the most recent surgery (laparoscopic ventral hernia repair with mesh)? |
| 16 | Vitals | What was the patient's condition like at the time of discharge, particularly focused on his vital signs, pain management and mobility? |
| 17 | Vitals | What was the range of the patient's blood pressure during her second hospital stay? |
| 18 | Vitals | What medication changes were made to the patient's blood pressure management during this hospital visit? |
| 19 | Test Results | Did the patient have any electrolyte imbalance during her first visit to the hospital? If yes, what were they? |
| 20 | Test Results | Did the patient's hemoglobin levels improve from 2166-11-22 to 2167-08-25? |
| 21 | Test Results | What was the fluctuation in the patient's presurgical bilirubin levels during his cholecystectomy procedure and how did it change post-surgery? |
| 22 | History | In the context of the alterations to the patient's medication list between the initial admission on 2188-12-29 and the subsequent admission on 2195-01-16, which of the following options accurately characterizes the modifications? |
| 23 | History | What is the patient's medical history and has there been a change in his medication over the course of two years from 2131-02-24 to 2133-01-20? |
| 24 | History | Has the patient experienced any previous hospital admissions or surgical interventions prior to the acute appendicitis and the associated laparoscopic appendectomy based on the discharge summary written on 2165-02-19? |
| 25 | Instruction | After the surgery, where was the patient instructed to put most of her weight and how long was she supposed to continue DVT prophylaxis using lovenox? |
| 26 | Instruction | How was the patient instructed on weight-bearing after his knee replacement? |
| 27 | Instruction | What is the advice given to the patient regarding weight bearing activity post-operation? |
| 28 | Plan | Was postoperative oral anticoagulation indicated for the patient, and if not, what justification is provided in the discharge summary for not prescribing it? |
| 29 | Plan | What was the result of the patient's EKG and what has been the treatment plan for his cardiac issues? |
| 30 | Plan | How was the patient's cellulitis condition managed during his hospital stay and what was his recommended treatment plan following discharge? |

# E   Model Evaluation

## E.1   GPT-4 Evaluation

For evaluating EHRNoteQA on open-ended and multi-choice formats, we utilized the `GPT4-turbo-preview (1106)` version with default hyperparameters, except for the temperature, which was set to 0. For answer generation (i.e., output) of each model, the temperature was set to 0 for both open-ended and multi-choice formats. The cost for evaluating the 962 QA data pairs was approximately $40 per model. The specific prompts for evaluating open-ended and multi-choice formats are shown in Figures 8 and 9. Given the prompt, the model output is evaluated as either correct or incorrect for both open-ended and multi-choice formats (1 for correct, 0 for incorrect).

We evaluate each model with GPT-4-Turbo 5 times and score each QA pair based on the mode value (either 1 or 0). When reporting the model scores in Table 4, we sum all the mode scores of each QA pair and convert the overall score to a 100-point scale. The reason for evaluating with the same prompt 5 times is that despite setting the temperature to 0, the same model output is scored differently as either correct or incorrect in some cases. Figure 11 shows the overall process of model output generation and evaluation for EHRNoteQA in open-ended and multi-choice formats.

To validate the accuracy of the mode-based evaluation, for open-ended format, three clinicians manually scored a total of 270 question-output pairs, which was compared to the evaluations by GPT-4-Turbo on the same set. Similar to the GPT-4-Turbo evaluation for open-ended format, clinicians were given the question, the patient discharge summaries, the correct answer, and the model output, and scored whether the model output is correct. For multi-choice format, the authors manually scored a total of 270 question-output pairs, which was compared to the evaluations by GPT-4-Turbo on the same set. Similar to the GPT-4-Turbo evaluation for multi-choice format, given the answer choices, the correct answer, and the model output, the authors scored whether the model output is correct.

We used the same GPT-based evaluation method for evaluating the discharge summary QA datasets (emrQA [44] and Yue et al. [65]), and the clinical domain knowledge benchmarks (MedQA [23], PubMedQA [24], MMLU* [19], MedMCQA [43])). For the discharge summary QA datasets, we utilized the prompt in Figure 8 if the QA pair included an answer; otherwise, we used the prompt in Figure 10 if the QA pair only consisted of evidence (*i.e.*, a text-span). For the clinical domain knowledge benchmarks, we utilized the prompt in Figure 9 as these benchmarks are in multi-choice formats.

**Open-Ended Evaluation (with Answer) Prompt**

Your task is to evaluate the provided model output in response to a specific question associated with the given discharge summaries. By using the correct answer also provided, you must score the answer as 0 or 1, based on the following scoring instructions.
Scoring Instructions:
1. Assign 1 point if the answer is correct.
2. Assign 0 points if the answer is either incorrect, or if it falsely claims there is no answer when one exists according to the discharge summaries.

- Discharge Summaries:
{{note}}

- Question: {{question}}

- Correct Answer:
{{answer}}

- Model Output:

{{output}}

Output format:
Score: {{score}}
Reasoning: {{explanation}}
(Note: In your response please replace score with the numerical score of 0 or 1, and provide a brief reasoning for the assigned score based on the evaluation criteria.)

Figure 8: Prompt Template for Open-Ended Evaluation with Answer

**Multi-Choice Evaluation Prompt**

Your task is to evaluate the provided model output by determining whether it matches the correct answer from the multiple-choice options provided. The model output is correct and should be met with a "yes" if it accurately reflects the content of the correct answer choice, not necessarily its exact wording. If the content of the model output aligns with the correct answer choice, despite any additional details or varied phrasing, you are to respond "yes". Should the model output diverge in meaning or substance from the correct answer—whether by selecting an alternative choice or providing a response not aligning with any provided options—a response of "no" is necessary.

Model Output:
{{output}}

Answer Choices:
{{choices}}

Correct Answer:
{{answer}}

With the given information, do you conclude that the model output substantively matches the correct answer provided? Respond solely with "yes" or "no".

Figure 9: Prompt Template for Multi-Choice Evaluation

---

**Open-Ended Evaluation (with Evidence) Prompt**

Your task is to evaluate the provided model output in response to a specific question associated with the given discharge summaries. By using the labeled evidence also provided, you must score the answer as 0 or 1, based on the following scoring instructions.
Scoring Instructions:
1. Assign 1 point if the answer is correct.
2. Assign 0 points if the answer is either incorrect, or if it falsely claims there is no answer when one exists according to the discharge summaries.

- Discharge Summaries:
`{{note}}`

- Question: `{{question}}`

- Labeled Evidence:
`{{evidence}}`

- Model Output:

`{{output}}`

Output format:
Score: `{{score}}`
Reasoning: `{{explanation}}`
(Note: In your response please replace score with the numerical score of 0 or 1, and provide a brief reasoning for the assigned score based on the evaluation criteria.)

---

Figure 10: Prompt Template for Open-Ended Evaluation with Evidence

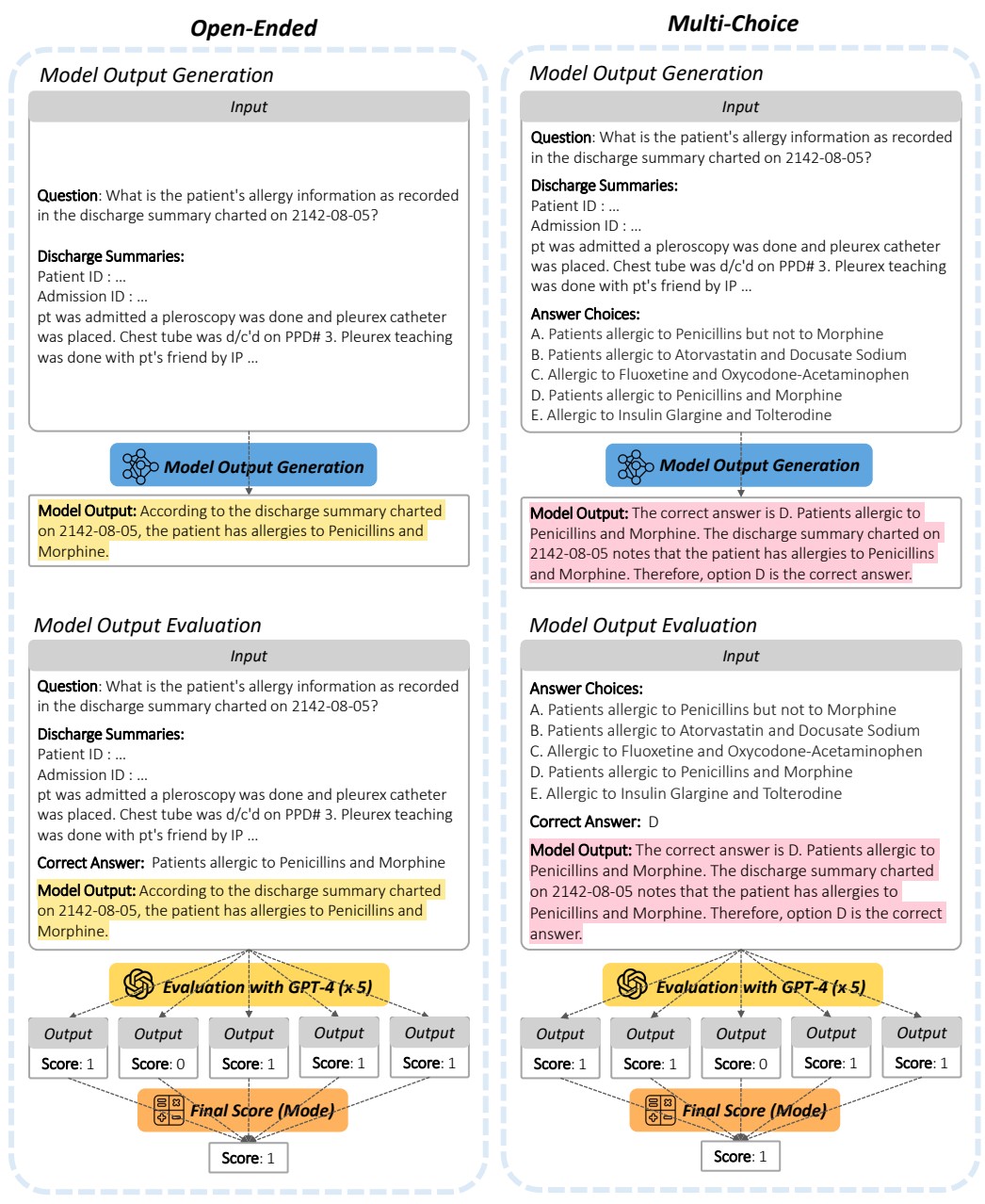

Figure 11: The overall process of model output generation and evaluation for EHRNoteQA in open-ended and multi-choice formats.

## E.2 LLMs performance by the number of notes

Table 10 shows the model performance across differing number of notes for Level 1 and Level 2 on both open-ended and multi-choice evaluation.

Table 10: Performance of 27 Large Language Models (LLMs) on EHRNoteQA using both multiple-choice and open-ended question answering methods. Empty cells indicate that the model does not support context lengths up to 8k for Level 2 data. [1]Asclepius is trained to provide only open-ended responses.

| Size | Model | Multi-Choice | | | | | Open-Ended | | | | | Foundation | Reference |
| | | Level 1 | | Level 2 | | | Level 1 | | Level 2 | | | | |
| | | 1 | 2 | 1 | 2 | 3 | 1 | 2 | 1 | 2 | 3 | | |
|---|---|---|---|---|---|---|---|---|---|---|---|---|---|
| | GPT4 | 96.97 | 97.36 | 96.55 | 97.22 | 91.67 | 95.08 | 87.55 | 86.90 | 84.03 | 88.89 | - | [42] |
| | GPT4-Turbo | 95.08 | 95.47 | 97.93 | 94.44 | 90.28 | 92.05 | 90.57 | 91.72 | 85.42 | 91.67 | - | [42] |
| | GPT3.5-Turbo | 89.77 | 86.79 | 87.59 | 87.50 | 79.86 | 84.47 | 80.00 | 81.38 | 76.39 | 68.75 | - | [6] |
| | Llama3-70b-Instruct | 94.32 | 94.34 | 95.17 | 91.67 | 88.89 | 91.29 | 86.79 | 90.34 | 81.94 | 88.19 | Llama3-70b | [1] |
| | Llama2-70b-chat | 87.50 | 82.26 | - | - | - | 80.68 | 76.98 | - | - | - | Llama2-70b | [56] |
| 70B | qCammel-70 | 89.02 | 82.26 | - | - | - | 78.79 | 77.74 | - | - | - | Llama2-70b | [55] |
| | Camel-Platypus2-70b | 90.91 | 88.68 | - | - | - | 81.44 | 76.23 | - | - | - | Llama2-70b | [28] |
| | Platypus2-70b-Instruct | 92.80 | 87.92 | - | - | - | 83.33 | 77.74 | - | - | - | Llama2-70b | [28] |
| 8x7B | Mixtral-8x7b-Instruct | 88.26 | 86.79 | 91.72 | 87.50 | 80.56 | 89.77 | 86.79 | 86.90 | 79.17 | 78.47 | Mistral-7b | [22] |
| 30B | MPT-30b-Instruct | 82.20 | 77.74 | 77.93 | 76.39 | 72.22 | 71.59 | 62.64 | 62.76 | 64.58 | 60.42 | MPT-30b-8k | [39] |
| | Llama2-13b-chat | 79.92 | 66.42 | - | - | - | 72.73 | 67.92 | - | - | - | Llama2-13b | [56] |
| | Vicuna-13b | 84.85 | 79.25 | - | - | - | 75.38 | 65.66 | - | - | - | Llama2-13b | [9] |
| | WizardLM-13b | 84.47 | 77.36 | - | - | - | 78.41 | 70.94 | - | - | - | Llama2-13b | [62] |
| 13B | qCammel-13 | 74.62 | 68.30 | - | - | - | 71.59 | 60.75 | - | - | - | Llama2-13b | [55] |
| | OpenOrca-Platypus2-13b | 89.77 | 82.26 | - | - | - | 81.82 | 76.60 | - | - | - | Llama2-13b | [29] |
| | Camel-Platypus2-13b | 80.68 | 75.47 | - | - | - | 71.97 | 63.77 | - | - | - | Llama2-13b | [28] |
| | Synthia-13b[1] | 82.58 | 75.85 | - | - | - | 76.52 | 72.45 | - | - | - | Llama2-13b | [54] |
| | Asclepius-13b[1] | - | - | - | - | - | 82.58 | 67.92 | - | - | - | Llama2-13b | [27] |
| | Gemma-7b-it | 79.92 | 75.09 | 72.41 | 67.36 | 61.81 | 72.35 | 55.09 | 66.90 | 53.47 | 42.36 | Gemma-7b | [52] |
| | MPT-7b-8k-instruct | 63.64 | 55.47 | 62.07 | 45.83 | 45.83 | 64.77 | 48.68 | 57.93 | 53.47 | 50.00 | MPT-7b-8k | [39] |
| | Mistral-7b-Instruct | 85.61 | 78.49 | 75.17 | 64.58 | 54.86 | 76.14 | 69.81 | 65.52 | 52.78 | 43.06 | Mistral-7b | [21] |
| | Dolphin-2.0-mistral-7b | 81.82 | 70.57 | - | - | - | 74.24 | 65.28 | - | - | - | Mistral-7b | [12] |
| 7B | Mistral-7b-OpenOrca | 89.02 | 85.28 | - | - | - | 82.58 | 76.60 | - | - | - | Mistral-7b | [31] |
| | SynthIA-7b | 82.95 | 73.96 | - | - | - | 76.52 | 72.83 | - | - | - | Mistral-7b | [53] |
| | Llama2-7b-chat | 71.21 | 60.38 | - | - | - | 65.91 | 52.08 | - | - | - | Llama2-7b | [56] |
| | Vicuna-7b | 83.33 | 73.21 | - | - | - | 66.67 | 52.83 | - | - | - | Llama2-7b | [9] |
| | Asclepius-7b[1] | - | - | - | - | - | 72.35 | 61.51 | - | - | - | Llama2-7b | [27] |

# F  Correlation Measurement

## F.1  DiSCQ Preprocessing

The DiSCQ [30] dataset is composed of 1,089 questions annotated by physicians. The questions are annotated based on a span of text (i.e., trigger) that prompted the question while reading the MIMIC-III discharge summaries [25]. An example of the triggering term and the corresponding question can be found below in Table 11. We have reformatted the original questions into the format *"With respect to Trigger, Question"*, following the approach of the DiSCQ authors. It's important to note that the dataset only provides the physicians' questions, without the corresponding answers.

Table 11: An example of a DiSCQ question and its reformatted version.

| | |
|---|---|
| **Discharge Summary** | CV : The patient 's family refused coronary artery catheterization . The patient was given ASA , Plavix , heparin drip x 24 hours , nitro drip , atorvastatin , metoprolol , and lisinopril . Her chest pain was controlled with morphine . Her SBP remained in the 160s-170s on hospital day 1 and she was gently diuresed . On hospital day 2 she experienced atrial fibrillation with HR in the 140s. Her metoprolol dose was increased from 25 mg PO bid to 50 mg PO bid ... |
| **Triggering Term** | atrial fibrillation |
| **Triggered Question** | Were interventions done? |
| **Reformatted Question** | With respect to atrial fibrillation, were interventions done? |

## F.2  Clinician Evaluation

From the 1,089 reformatted DiSCQ questions, we randomly sampled 300 questions and assigned 100 each to three clinicians. Table 12 shows the indices of the DiSCQ data that were assigned to each clinician. These indices correspond to the row indices in the file `discq_questions_final.csv`, provided by the DiSCQ dataset[11]. Each clinician was tasked to assess the output of 19 models as either correct or incorrect, for their assigned 100 questions (totaling 1,900 model outputs per clinician). To avoid bias, the order of model answers given to the clinicians was shuffled for each question. Similar to Appendix C.3, we used Streamlit and Google Sheets for clinician evaluations, with the screenshots shown in Figures 12 and 13.

Table 12: Row index of the DiSCQ questions in `discq_questions_final.csv` given to each clinician for evaluation

| Clinician | Index |
|---|---|
| A | 519, 0, 276, 238, 537, 62, 775, 716, 439, 334, 407, 43, 275, 319, 413, 23, 45, 267, 162, 418, 880, 203, 210, 456, 823, 1011, 202, 778, 4, 15, 116, 802, 106, 222, 234, 815, 435, 768, 2, 383, 467, 546, 402, 509, 707, 939, 385, 720, 508, 1019, 47, 801, 94, 794, 205, 104, 393, 394, 65, 73, 68, 564, 514, 1013, 91, 67, 957, 905, 702, 357, 534, 297, 254, 64, 452, 374, 408, 410, 147, 825, 289, 736, 830, 415, 220, 436, 117, 420, 271, 1033, 984, 382, 265, 433, 182, 934, 788, 741, 985, 840 |
| B | 582, 341, 1057, 1074, 457, 548, 870, 783, 26, 952, 714, 100, 931, 797, 1020, 958, 597, 543, 51, 475, 989, 583, 838, 204, 152, 127, 464, 280, 8, 422, 30, 82, 180, 195, 307, 315, 1083, 911, 346, 700, 326, 827, 1010, 795, 78, 89, 388, 1053, 330, 291, 871, 358, 1015, 841, 527, 263, 292, 738, 14, 96, 535, 183, 164, 155, 704, 1042, 852, 441, 839, 54, 1043, 58, 781, 160, 432, 75, 517, 1035, 466, 740, 708, 264, 40, 787, 39, 566, 5, 513, 1002, 739, 846, 258, 332, 367, 79, 361, 114, 938, 994, 340 |
| C | 595, 344, 1070, 1076, 465, 567, 876, 800, 27, 954, 732, 98, 935, 817, 1021, 980, 686, 563, 52, 485, 987, 596, 848, 200, 130, 118, 476, 279, 7, 428, 29, 87, 158, 190, 303, 313, 1084, 932, 348, 709, 329, 842, 1012, 816, 83, 90, 390, 1054, 335, 288, 877, 360, 1017, 851, 539, 259, 290, 747, 9, 95, 547, 161, 156, 148, 713, 1051, 875, 450, 849, 55, 1052, 59, 798, 153, 438, 80, 528, 1039, 477, 765, 718, 260, 38, 803, 37, 587, 6, 524, 1003, 748, 872, 257, 336, 368, 84, 363, 111, 942, 995, 343 |

---

[11] https://physionet.org/content/discq/1.0/

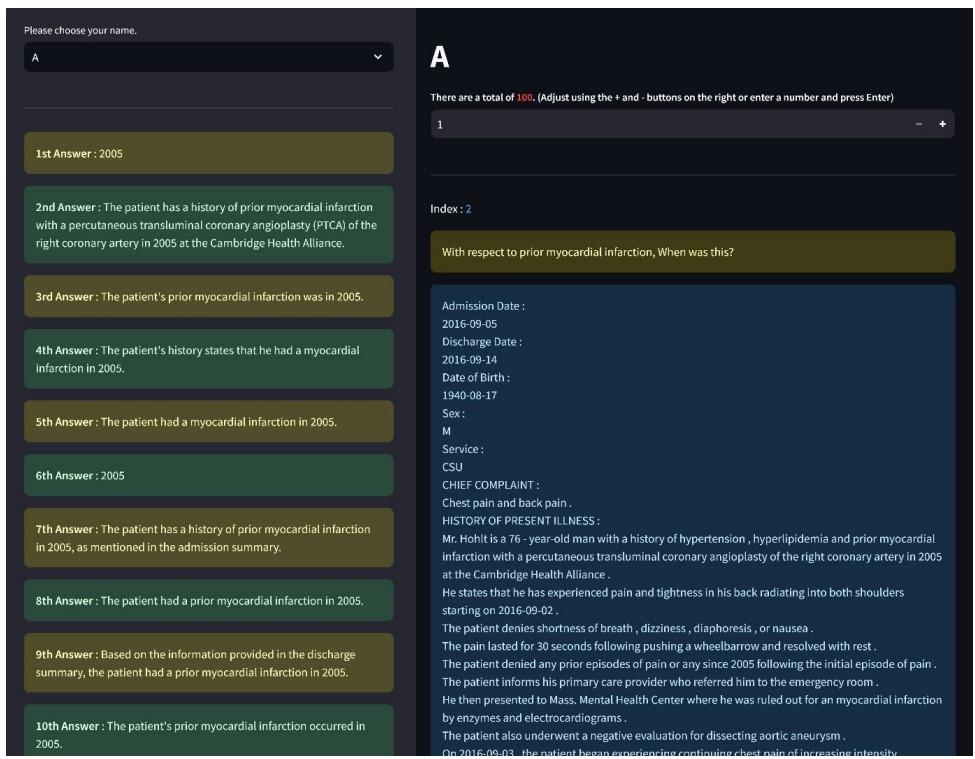

Figure 12: Screenshot of Streamlit provided to clinicians for DiSCQ model response evaluation. For each question, the clinicians reviewed the outputs of 19 responses, and assessed the output of each model as either correct or incorrect.

| | Model 1 | Model 2 | Model 3 | Model 4 | Model 5 | Model 6 | Model 7 | Mo |
|---|---|---|---|---|---|---|---|---|
| 1 | o | o | o | x | o | o | x | |
| 2 | x | o | o | o | o | o | x | |
| 3 | o | o | o | o | o | o | o | |
| 4 | o | x | o | o | o | o | o | |
| 5 | x | x | x | o | o | o | x | |
| 6 | o | o | o | o | o | o | o | |
| 7 | x | o | o | x | o | x | x | |
| 8 | o | x | o | o | o | o | x | |
| 9 | x | x | x | x | o | x | o | |
| 10 | o | o | o | o | o | o | o | |
| | | | | | | | | |
| 11 | o | x | o | o | o | o | o | |

Figure 13: Screenshot of Google Sheet provided to clinicians for DiSCQ model response evaluation. The clinicians marked the model outputs of each question as either correct (o) or incorrect (x).

## F.3 Model Performance on Benchmarks

Tables 13 and 14 show the model scores of DiSCQ evaluated by 3 clinicians, along with the scores of different benchmarks including EHRNoteQA. The intra-clinician correlation and the benchmark correlation results in Table 5 are calculated based on the values in Tables 13 and 14.

Table 15 shows the model scores of evaluating EHRNoteQA using different evaluation methods. The correlation results regarding different evaluation methods in Table 5 are calculated based on the values in Table 15 with the 3 clinician scores in Table 13.

Table 13: Model scores of DiSCQ questions evaluated by 3 clinicians, EHRNoteQA (both open-ended and multi-choice), and the two discharge summary QA datasets (emrQA [44] and Yue et al. [65]).

| Model | Clinician A | Clinician B | Clinician C | EHRNoteQA (Open-Ended) | EHRNoteQA (Multi-Choice) | emrQA | Yue et al., |
|---|---|---|---|---|---|---|---|
| Llama2-70b-chat | 37 | 53 | 48 | 78.83 | 84.88 | 65.59 | 70.09 |
| qCammel-70 | 52 | 45 | 60 | 78.26 | 85.63 | 61.05 | 76.64 |
| Camel-Platypus2-70b | 68 | 62 | 69 | 78.83 | 89.79 | 67.1 | 77.57 |
| Platypus2-70b-Instruct | 77 | 75 | 88 | 80.53 | 90.36 | 76.55 | 69.16 |
| MPT-30b-Instruct | 22 | 28 | 41 | 67.11 | 79.96 | 67.29 | 67.5 |
| Llama2-13b-chat | 45 | 57 | 52 | 70.32 | 73.65 | 53.3 | 54.21 |
| Vicuna-13b | 56 | 60 | 61 | 70.51 | 82.04 | 74.66 | 61.68 |
| WizardLM-13b | 57 | 58 | 65 | 74.67 | 80.91 | 70.69 | 68.22 |
| qCammel-13 | 40 | 31 | 45 | 66.16 | 71.46 | 57.46 | 44.86 |
| OpenOrca-Platypus2-13b | 65 | 69 | 76 | 79.21 | 86.01 | 75.42 | 73.83 |
| Camel-Platypus2-13b | 49 | 40 | 57 | 67.86 | 78.07 | 59.92 | 47.66 |
| Synthia-13b | 53 | 51 | 52 | 74.48 | 79.21 | 69.75 | 67.29 |
| Llama2-7b-chat | 17 | 38 | 33 | 58.98 | 65.78 | 57.46 | 57.94 |
| Vicuna-7b | 52 | 42 | 54 | 59.74 | 78.26 | 63.51 | 45.79 |
| Mistral-7b-Instruct | 55 | 55 | 59 | 72.97 | 82.04 | 76.74 | 63.55 |
| MPT-7b-8k-instruct | 23 | 28 | 20 | 56.71 | 59.55 | 60.3 | 61.68 |
| Dolphin-2.0-mistral-7b | 51 | 55 | 57 | 69.75 | 76.18 | 75.42 | 77.57 |
| Mistral-7b-OpenOrca | 67 | 61 | 66 | 79.58 | 87.15 | 78.63 | 78.5 |
| SynthIA-7b | 55 | 51 | 58 | 74.67 | 78.45 | 65.59 | 78.5 |

Table 14: Model scores of clinical domain knowledge benchmarks (MedQA [23], PubMedQA [24], MMLU* [19], MedMCQA [43])) and general benchmarks (ARC [10], HellaSwag [66], MMLU [19], TruthfulQA [35] ,Winogrande [49], GSM8K [11]).

| Model | MedQA | PubMedQA | MMLU* | MedMCQA | ARC | HellaSwag | MMLU | TruthfulQA | Winogrande | GSM8k | AVG |
|---|---|---|---|---|---|---|---|---|---|---|---|
| Llama2-70b-chat | 44.38 | 67.8 | 58.7 | 37.84 | 64.59 | 85.88 | 63.91 | 52.8 | 80.51 | 26.69 | 62.40 |
| qCammel-70 | 55.77 | 62.6 | 66.77 | 43.61 | 68.34 | 87.87 | 70.18 | 57.47 | 84.29 | 29.72 | 66.31 |
| Camel-Platypus2-70b | 60.02 | 64 | 71.7 | 49.8 | 71.08 | 87.6 | 70.04 | 58.09 | 83.82 | 22.9 | 65.59 |
| Platypus2-70b-Instruct | 56.17 | 64.2 | 73.79 | 50.59 | 71.84 | 87.94 | 70.48 | 62.26 | 82.72 | 40.56 | 69.30 |
| MPT-30b-Instruct | 41.01 | 68 | 50.31 | 39.06 | 58.45 | 84.31 | 49.15 | 38.05 | 75.14 | 15.31 | 53.40 |
| Llama2-13b-chat | 36.61 | 58.8 | 49.16 | 32.92 | 59.04 | 81.94 | 54.64 | 44.12 | 74.51 | 15.24 | 54.92 |
| Vicuna-13b | 43.21 | 66.8 | 57.86 | 40.21 | 57.08 | 81.24 | 56.67 | 51.51 | 74.66 | 11.3 | 55.41 |
| WizardLM-13b | 39.75 | 57.2 | 51.99 | 33.99 | 59.04 | 82.21 | 54.64 | 47.27 | 71.9 | 13.5 | 54.76 |
| qCammel-13 | 41.48 | 55.8 | 51.68 | 33.66 | 60.84 | 83.66 | 56.73 | 47.54 | 76.16 | 11.37 | 56.05 |
| OpenOrca-Platypus2-13b | 44.3 | 60.6 | 59.12 | 43.1 | 62.8 | 83.15 | 59.39 | 53.08 | 76.24 | 9.02 | 57.28 |
| Camel-Platypus2-13b | 46.66 | 61 | 59.12 | 39.3 | 60.75 | 83.61 | 56.51 | 49.6 | 75.37 | 0.08 | 54.32 |
| Synthia-13b | 40.61 | 54.8 | 51.47 | 38.32 | 61.26 | 82.93 | 56.47 | 47.27 | 76.48 | 10.99 | 55.90 |
| Llama2-7b-chat | 35.43 | 58.6 | 44.23 | 31.8 | 52.9 | 78.55 | 48.32 | 45.57 | 71.74 | 7.35 | 50.74 |
| Vicuna-7b | 38.65 | 63.4 | 49.37 | 35.29 | 53.24 | 77.39 | 51.04 | 50.34 | 72.14 | 8.19 | 52.06 |
| Mistral-7b-Instruct | 41.16 | 51 | 55.45 | 38.99 | 54.52 | 75.63 | 55.38 | 56.28 | 73.72 | 14.25 | 54.96 |
| MPT-7b-8k-instruct | 33.62 | 59.1 | 38.99 | 35.67 | 45.9 | 74.47 | 41.97 | 35.21 | 65.98 | 20.7 | 47.37 |
| Dolphin-2.0-mistral-7b | 43.99 | 52.4 | 56.5 | 38.58 | 59.22 | 80.26 | 56.9 | 61.09 | 75.37 | 18.65 | 58.58 |
| Mistral-7b-OpenOrca | 47.29 | 62 | 60.06 | 40.23 | 64.08 | 83.99 | 62.24 | 53.05 | 77.74 | 19.94 | 60.17 |
| SynthIA-7b | 47.13 | 69 | 53.77 | 42.67 | 62.12 | 83.45 | 62.65 | 51.37 | 78.85 | 17.59 | 59.34 |

Table 15: Model scores of evaluating EHRNoteQA using GPT-4 (ours) for both multi-choice and open-ended, probability-based for multi-choice, and BLEU and ROUGE-L for open-ended formats.

| Model | Multi-Choice(GPT4) | Multi-Choice(Prob-Index) | Multi-Choice(Prob-Value) | Open-Ended(GPT4) | Open-Ended(BLEU) | Open-Ended(ROUGE-L) |
|---|---|---|---|---|---|---|
| Llama2-70b-chat | 84.88 | 82.99 | 73.53 | 78.83 | 0.058899 | 0.254175 |
| qCammel-70 | 85.63 | 88.09 | 72.59 | 78.26 | 0.058725 | 0.251194 |
| Camel-Platypus2-70b | 89.79 | 87.71 | 76.75 | 78.83 | 0.065076 | 0.262653 |
| Platypus2-70b-Instruct | 90.36 | 89.22 | 80.15 | 80.53 | 0.078661 | 0.285784 |
| MPT-30b-Instruct | 79.96 | 77.88 | 73.16 | 67.11 | 0.084820 | 0.261259 |
| Llama2-13b-chat | 73.65 | 71.83 | 65.41 | 70.32 | 0.045917 | 0.232470 |
| Vicuna-13b | 82.04 | 78.64 | 69.57 | 70.51 | 0.058815 | 0.259319 |
| WizardLM-13b | 80.91 | 76.18 | 68.62 | 74.67 | 0.050528 | 0.234151 |
| qCammel-13 | 71.46 | 76.37 | 61.44 | 66.16 | 0.052681 | 0.231925 |
| OpenOrca-Platypus2-13b | 86.01 | 82.99 | 72.21 | 79.21 | 0.057719 | 0.250524 |
| Camel-Platypus2-13b | 78.07 | 71.64 | 62 | 67.86 | 0.043034 | 0.215407 |
| Synthia-13b | 79.21 | 77.69 | 70.32 | 74.48 | 0.081454 | 0.289530 |
| Llama2-7b-chat | 65.78 | 59.17 | 56.52 | 58.98 | 0.044326 | 0.220668 |
| Vicuna-7b | 78.26 | 71.08 | 65.6 | 59.74 | 0.054364 | 0.246746 |
| Mistral-7b-Instruct | 82.04 | 72.97 | 63.14 | 72.97 | 0.071142 | 0.281530 |
| MPT-7b-8k-instruct | 59.55 | 49.53 | 53.88 | 56.71 | 0.085256 | 0.259039 |
| Dolphin-2.0-mistral-7b | 76.18 | 78.45 | 63.33 | 69.75 | 0.073931 | 0.279292 |
| Mistral-7b-OpenOrca | 87.15 | 84.69 | 71.27 | 79.58 | 0.084700 | 0.294611 |
| SynthIA-7b | 78.45 | 83.18 | 71.27 | 74.67 | 0.073345 | 0.278483 |

