# OpenReview forum: "EHRNoteQA: An LLM Benchmark for Real-World Clinical Practice Using Discharge Summaries"
_NeurIPS.cc/2024/Datasets_and_Benchmarks_Track — NeurIPS 2024 Track Datasets and Benchmarks Poster_

### Official Review · Reviewer_XMWc · 2024-07-02
**This work presents the EHRNoteQA dataset for evaluating LLMs for clinical practice Using discharge summaries**

**Rating:** 6
**Confidence:** 4
**Clarity:** The paper is well-written and organized.

**Review:**

Please see the strengths and opportunities for improvement for further review details.

**Strengths:**

- With the advent of LLMs and their applications in the clinical domain, there is a need to develop a benchmark that helps in assessing their real-life clinical applicability. This benchmark serves this purpose to some extent.

- Evaluating LLMs’ long-context understanding is an important area and this benchmark can help in this direction since it involves questions where you need to connect information through multiple discharge summaries to answer.

- EHRNoteQA also covers more topics (~8) modeling data closer to real-world settings.

- Experiments are comprehensive (evaluation of 27 LLMs) and the paper is well-written.

**Additional Feedback:**

N/A

**Correctness:**

Please see the "opportunities for improvements" section for the comments related to evaluation metrics and experimental setup.

**Documentation:**

Yes

**Limitations:**

Yes, the authors have adequately addressed the limitations and potential negative societal impact of their work.

**Opportunities For Improvement:**

- The authors have used GPT-4 to generate <question, answer> pairs based on discharge summaries, including incorrect options. According to the provided statistics, the authors have generated 962 questions for 962 distinct patients, resulting in one question per patient per topic. Why not generate one question per topic for each patient, leading to a dataset with a total of (962 x 8) <question, answer> pairs? This approach could enhance the data size, diversity, and comprehensiveness of the benchmark. Is there a specific reason for the chosen approach of generating only one question per patient in the paper?

- The authors evaluated 27 models by providing <question, discharge summaries> pairs directly to the model as input (shown in Figure 11 for both settings). However, why were no additional task instructions provided? Evaluating the models' performance in a Chain-of-Thought (CoT) setting is becoming a de facto method, as it helps to understand the models’ ability to reach answers based on correct rationale and provides greater insights into their performance. Why was CoT or similar methods not used for this task? Is there a specific reason for this choice?

- In section 4.1, the authors used GPT-4 as the evaluator for open-ended questions-answering, however, it is helpful if authors also present some objective metrics such as exact-match (EM) or similarity-based metrics (such as cosine similarity) to support the results obtained using GPT-4. Furthermore, what is the accuracy of GPT-4 in evaluating open-ended questions-answering?

- In addition, why authors have used GPT-4 to evaluate the MCQA setting? There could be simple accuracy can be presented since the correct choice is known as the gold answer.

- In section 4.1, it can be helpful to provide the temperature value at which these models are evaluated. Is the inference setting for these LLMs deterministic? Also, it can be helpful to provide temperature values used for GPT-4 during data generation as well as evaluation.

- From Table 3, we can see that GPT-4 archives ~97% and ~95% performance on level 1 and level 2, respectively. Also, the open-source model Llama-3-70b-Instruct archives ~94% and ~92% performance on level 1 and level 2, respectively. Here, it is hard to grasp the usefulness of the dataset and understand the challenges presented since these models are achieving more than 90% performance by just providing instances as input. GPT-4 could further improve the performance on this benchmark if a different prompt setting is provided. Therefore, it is essential to include a discussion that clearly outlines the challenges in this benchmark for the research community. This discussion could highlight the difficulties in the dataset that go beyond achieving high accuracy, emphasizing areas where current models still need improvement.

- In section 4.2, the authors present the findings from Table 3 but do not provide a detailed explanation of the reasons behind these findings. I think a CoT kind of experimental setting (by exploring how models arrive at their answers) could help understand such models' behaviors. Please add some detailed analysis related to these findings.

**Relation To Prior Work:**

Related work is thorough, however, I think having a one comparison table with existing datasets in terms of various dataset features such as (not limited to) topics covered, and complexity of questions can improve the comprehensiveness of the section.

**Summary And Contributions:**

This paper introduced the EHRNoteQA, a systematically created clinician-validated question-answering benchmark with two tasks: (1) open-ended, and (2) MCQA based on one/multiple discharge summaries. This benchmark has two key properties compared to previous datasets: (1) covers eight different topics, and (2) includes questions that need multiple discharge summaries to answer. Also, this paper proposes a reliable evaluation method for each task. Furthermore, this work evaluates 27 different LLMs and presents various challenges.

---

> ### Author Rebuttal · Authors · 2024-08-15
>
> Thank you for your thoughtful and constructive reviews.
>
> From your feedback, we believe you place significant value on the efficacy of benchmarks in assessing model performance. We share this belief, recognizing that benchmarks are essential for performance comparison and improving LLMs.
>
> However, our primary goal in this work focuses on how these benchmarks connect to real-world applications. If Model A achieves higher scores than Model B on datasets such as MMLU or ARC, what guarantees do we have regarding its performance in real use-cases? How can these results be practically leveraged?
>
> Our approach with EHRNoteQA focuses on reflecting real-world scenarios rather than serving as a comparative tool for model performances. By doing so, we aim to bridge the gap between benchmark tests and genuine clinical applications. This approach ensures that the models we evaluate and develop are deeply relevant and effective in practical, real-world clinical settings.
> We would like to discuss this aspect further to clarify EHRNoteQA’s position in terms of real-world applicability.
>
> Thank you once again for your valuable feedback. We hope this perspective offers a comprehensive understanding of our intentions and the real-world significance of EHRNoteQA.

---

> > ### Author Response · Authors · 2024-08-25
> > **Sincerely expecting further discussions with Reviewer XMWc**
> >
> > Dear Reviewer XMWc,
> >
> > As the discussion period is nearing its end, we would like to inquire if our response addressed your concerns.
> >
> > Please let us know if you have any further questions or feedback.

---

> > > ### Comment · Reviewer_XMWc · 2024-08-27
> > >
> > > Overall majority of my questions were answered, however, there are some small concerns that I responded to in the respective responses. Considering overall efforts in the rebuttal, I am inclined towards increasing my score.

---

> > > > ### Author Response · Authors · 2024-08-30
> > > >
> > > > Thank you for your valuable comments. We have carefully reviewed each of your points and provided our responses above. We hope our responses clarify your comments, and look forward to any further discussion.

---

> ### Author Rebuttal · Authors · 2024-08-15
>
> **Q7.**
> Related work is thorough, however, I think having a one comparison table with existing datasets in terms of various dataset features such as (not limited to) topics covered, and complexity of questions can improve the comprehensiveness of the section.
>
> **A7.**
> We agree that including a comparison table in the related works section would enhance the comprehensiveness of our paper. Based on your suggestions, we will add the table below in the camera-ready version of our paper.
>
> | **Dataset**              | **Questions** | **Documents** | **Patients** | **Data Source**                        | **Answer Format**             | **Single/Multiple Documents** | **Topics** |
> |--------------------------|-----------------|-----------------|----------------|----------------------------------------|-------------------------------|--------------------------------|--------------|
> | Pampari et al. (2018)     | 73,111          | 303             | 303            | Discharge Summaries (n2c2)             | Text Span                     | Single                         | 5            |
> | Fan (2019)                | 245             | 138             | 138            | Discharge Summaries (n2c2)             | Text Span                     | Single                         | 1            |
> | Yue et al. (2021)         | 1,287           | 36              | 36             | Clinical Notes (MIMIC-III)             | Text Span                     | Single                         | 5            |
> | Moon et al. (2023)        | 96,939          | 505             | 505            | Discharge Summaries (n2c2)             | Text Span                     | Single                         | 1            |
> | **EHRNoteQA (ours)**      | **962**         | **1,659**       | **962**        | **Discharge Summaries (MIMIC-IV)**     | **Multi-Choice & Open-Ended** | **Multiple**                   | **8**        |

---

> ### Author Rebuttal · Authors · 2024-08-15
>
> **Q6.**
> From Table 3, we can see that GPT-4 archives ~97% and ~95% performance on level 1 and level 2, respectively. Also, the open-source model Llama-3-70b-Instruct archives ~94% and ~92% performance on level 1 and level 2, respectively. Here, it is hard to grasp the usefulness of the dataset and understand the challenges presented since these models are achieving more than 90% performance by just providing instances as input. GPT-4 could further improve the performance on this benchmark if a different prompt setting is provided. Therefore, it is essential to include a discussion that clearly outlines the challenges in this benchmark for the research community. This discussion could highlight the difficulties in the dataset that go beyond achieving high accuracy, emphasizing areas where current models still need improvement.
>
> **A6.**
> Thank you for highlighting this crucial point. It’s true that models such as GPT-4 and LLaMA-70b-Instruct already demonstrate high performance, with accuracy levels surpassing 90%. This naturally raises questions about the complexity of the tasks within EHRNoteQA and whether the dataset offers sufficient challenges to justify its use.
>
> However, the primary objective of our benchmark is not merely to present complex problems that push models to their limits just to set a new state-of-the-art. Instead, our focus is on reflecting real-world, clinically relevant questions—those that clinicians encounter in practice—and providing a benchmark that closely aligns with the evaluations clinicians would make.
>
> We acknowledge that the high performance of these models on our dataset might suggest a reduced need for such a dataset. However, intentionally crafting more complex or artificially challenging questions to lower model performance would go against the core purpose of our research. Our goal is to mirror the practical utility of these models in a clinical setting, not to artificially increase the level of difficulty.
>
> Furthermore, it's important to consider the practical constraints within real-world hospital environments. Due to privacy concerns, API-based models such as GPT and Claude cannot easily be deployed in actual clinical settings. While HIPAA-compliant platforms are emerging, de-identification and secure handling of patient records remain significant challenges. Additionally, many hospitals—especially those without the resources of major institutions—are unable to run models as large as 70b parameters and instead must rely on models that can operate within their limited GPU capabilities.
>
> In this context, our dataset is highly valuable. It allows hospitals to select LLMs that perform well under these constraints. Our experiment results show that EHRNoteQA is more aligned with actual clinical evaluations compared to other benchmarks such as MMLU or MedQA (see Section 4.3). Therefore, EHRNoteQA stands as the most relevant benchmark for assessing and selecting LLMs for deployment in real-world clinical settings.

---

> ### Author Rebuttal · Authors · 2024-08-15
>
> **Q5.**
> In section 4.1, it can be helpful to provide the temperature value at which these models are evaluated. Is the inference setting for these LLMs deterministic? Also, it can be helpful to provide temperature values used for GPT-4 during data generation as well as evaluation.
>
> **A5.**
> For model evaluation using GPT-4, the temperature was set to 0 (noted in Appendix E.1). For the generation of the EHRNoteQA dataset with GPT-4, the temperature was set to 1 (noted in Appendix C.2). For model inference (i.e., model output generation), the temperature was set to 0. Thank you for pointing this out and we will ensure that the temperature value for model inference is also clearly included in the camera-ready version of our paper.

---

> ### Author Rebuttal · Authors · 2024-08-15
>
> **Q4.**
> In addition, why authors have used GPT-4 to evaluate the MCQA setting? There could be simple accuracy can be presented since the correct choice is known as the gold answer.
>
> **A4.**
> We recognize that simpler methods, such as regular expressions or direct matching techniques, could be used to evaluate MCQA since the correct answer is known. However, we chose to use GPT-4 for evaluation due to the variability in model outputs, especially in zero-shot settings (which was an inevitable choice due to current LLMs’ context length constraints) where responses can differ significantly. For example, outputs might range from straightforward answers like "The answer is A.", “[Reasoning], therefore A is the correct answer.” or just "A," to more implicit responses such as "The patient underwent pipeline embolization," where the corresponding answer choice is not explicitly stated. Some outputs may also include reasoning, such as "Choice B is not the answer because...," making it difficult to rely on fixed-rule evaluations.
>
> Given these challenges, we used GPT-4 for the MCQA evaluation. To ensure reliability, we manually verified GPT-4’s MCQA evaluation, which showed 98% accuracy (as noted in Section 4.1). We also experimented with probability-based evaluations using tools like LM-Evaluation-Harness (https://github.com/EleutherAI/lm-evaluation-harness), which are commonly used in MCQA scoring. However, these evaluations did not align as well with clinician-evaluated LLM scores as GPT-4 did. Based on your feedback, we will clarify the rationale for using GPT-4 for MCQA evaluation in the camera-ready version to ensure there is no confusion for readers.

---

> > ### Comment · Reviewer_XMWc · 2024-08-27
> >
> > I am not sure if that's the limitation, since you can always specify the format in the prompt. For example, you can say `please use the format while answering: [A/B/C/D]` or something similar and then parses it.

---

> > > ### Author Rebuttal · Authors · 2024-08-30
> > >
> > > We appreciate your suggestion regarding the use of explicit instructions in the prompt, such as specifying the format, and then parsing the output accordingly. In fact, we initially considered and experimented with this approach ourselves.
> > >
> > > During our experiments, we observed that while models such as GPT-4, which are meticulously fine-tuned (alignment-tuned) for instruction-following, generally adhere to these strict instructions (though not perfectly—it's worth noting that even GPT-4 doesn't achieve a perfect score on the instruction-following benchmark, IF eval[1]), most open-source models do not consistently follow these instructions.
> > >
> > > For instance, even when we used prompts such as "please use the format while answering: [A/B/C/D]," the responses from models often deviated from the desired format. Models produced outputs such as :
> > > * "The answer is A. The patient was fairly stabilized, with pain under control, consuming a regular diet, and able to walk and relieve himself without assistance."
> > > * "A. The patient was fairly stabilized, with pain under control, consuming a regular diet, and able to walk and relieve himself without assistance."
> > > * “A is the correct answer.”
> > >
> > > Additionally, some models provided further explanations, such as, "The answer B is not correct because...," which does not conform to the specified format. If we were to evaluate such responses using regex, based on the exact format provided in the prompt, many of the answers would be marked as incorrect, even if they contained the correct answer. This would result in an evaluation that prioritizes adherence to instructions over the accuracy of the answer in the context of EHRNoteQA.
> > >
> > > We agree that regex-based evaluation could simplify the process and make it more straightforward for users of the benchmark. However, due to the current limitations and variability in instruction-following capabilities among different models, we decided to use GPT-4 for evaluation, which we empirically found higher evaluation accuracy compared to regex, to enable more fair and accurate assessment across all models.
> > >
> > > However, as language models continue to improve, we anticipate that adherence to instructions will become more reliable. We are committed to revisiting and potentially updating our evaluation methods for EHRNoteQA and LLM scores, adopting a simpler approach in the future.
> > >
> > > [1] Zhou et al., "Instruction-following evaluation for large language models" arXiv preprint arXiv:2311.07911 (2023).

---

> ### Author Rebuttal · Authors · 2024-08-15
>
> **Q3.**
> In section 4.1, the authors used GPT-4 as the evaluator for open-ended questions-answering, however, it is helpful if authors also present some objective metrics such as exact-match (EM) or similarity-based metrics (such as cosine similarity) to support the results obtained using GPT-4. Furthermore, what is the accuracy of GPT-4 in evaluating open-ended questions-answering?
>
> **A3.**
> Based on your feedback, we have also incorporated **exact match scores** and **semantic similarity** into our evaluation metrics. For Semantic Similarity, we utilized two models: sentenceBERT and clinicalBERT, measuring the cosine similarity between the embeddings of the model output and the gold standard answers. The results of these additional evaluations are presented below.
>
>
> | Model                           | EM (F1 score) | Cosine (SentenceBERT) | Cosine (Clinical BERT) |
> |---------------------------------|---------------|-----------------------|------------------------|
> | Llama2-70b-chat                 | 0.2777        | 0.6840                | 0.9321                 |
> | qCammel-70                      | 0.2684        | 0.6804                | 0.9238                 |
> | Camel-Platypus2-70b             | 0.2851        | 0.6878                | 0.9285                 |
> | Platypus2-70b-Instruct          | 0.3177        | 0.6944                | 0.9364                 |
> | MPT-30b-Instruct                | 0.3016        | 0.6357                | 0.9125                 |
> | Llama2-13b-chat                 | 0.2596        | 0.6674                | 0.9281                 |
> | Vicuna-13b                      | 0.2803        | 0.6679                | 0.9278                 |
> | WizardLM-13b                    | 0.2536        | 0.6679                | 0.9273                 |
> | qCammel-13                      | 0.2495        | 0.6624                | 0.9205                 |
> | OpenOrca-Platypus2-13b          | 0.2702        | 0.6679                | 0.9198                 |
> | Camel-Platypus2-13b             | 0.2302        | 0.6535                | 0.9167                 |
> | Synthia-13b                     | 0.3184        | 0.6833                | 0.9341                 |
> | Llama2-7b-chat                  | 0.2452        | 0.6634                | 0.9244                 |
> | Vicuna-7b                       | 0.2684        | 0.6599                | 0.9284                 |
> | Mistral-7b-Instruct             | 0.3072        | 0.6829                | 0.9357                 |
> | MPT-7b-8k-instruct              | 0.2979        | 0.6280                | 0.9174                 |
> | Dolphin-2.0-mistral-7b          | 0.3055        | 0.6637                | 0.9301                 |
> | Mistral-7b-OpenOrca             | 0.3280        | 0.6991                | 0.9399                 |
> | SynthIA-7b                      | 0.3021        | 0.6856                | 0.9336                 |
>
>
> We measured the correlation between clinician evaluations and the exact-match  and similarity-based metric results respectively, which are shown below. The GPT-4, BLEU, and ROUGE-L correlations are from Table 4 of our paper.
>
>
> |  | Clinician A |  | Clinician B |  | Clinician C |  |
> |--------|-------------|-------------|-------------|-------------|-------------|-------------|
> |  | Spearman | Kendall | Spearman | Kendall | Spearman | Kendall |
> | **GPT-4**        | **0.770**       | **0.609**    | **0.805**       | **0.617**    | **0.801**       | **0.657**    |
> | **BLEU**         | 0.155       | 0.112    | 0.037       | 0.059    | 0.014       | -0.006   |
> | **ROUGE-L**      | 0.500       | 0.324    | 0.398       | 0.283    | 0.356       | 0.241    |
> | **EM (F1 score)**| 0.422       | 0.288    | 0.336       | 0.236    | 0.266       | 0.194    |
> | **Cosine (SentenceBERT)** | 0.710  | 0.524    | 0.726       | 0.555    | 0.652       | 0.453    |
> | **Cosine (Clinical BERT)** | 0.536  | 0.382    | 0.552       | 0.389    | 0.394       | 0.288    |
>
>
> While using GPT-4 still exhibited the highest correlation with clinician evaluations, the semantic similarity method—particularly using sentenceBERT for cosine similarity—demonstrated the next highest correlation. We will make sure to incorporate the exact match and cosine similarity model scores and correlations in the camera-ready version of our paper, along with the GPT-4, BLEU, ROUGE-L model scores and correlation values.
>
> In evaluating GPT-4’s performance on open-ended questions, we used Cohen's kappa agreement (see Section 4.2) rather than the accuracy metric. While measuring the accuracy of GPT-4 is suitable for multiple-choice QA, where model outputs can be directly compared to a gold answer, this approach does not apply to open-ended QA. In open-ended QA scenarios, clinicians must manually assess the model's responses, and these clinician assessments should be compared with the GPT-4 evaluations. Given the subjective nature of open-ended questions, even with a gold-standard answer, there is inherent variability in human evaluations, as reflected in the intra-rater agreement values in Section 4.2. To address this, we measured Cohen’s kappa agreement among clinicians and Cohen's kappa agreement between evaluations of each clinician and GPT-4. The findings indicate that the agreement between clinicians and GPT-4 is within the range of clinician agreement, supporting the validity of using GPT-4 for evaluating open-ended QA.

---

> ### Author Rebuttal · Authors · 2024-08-15
>
> **Q2.**
> The authors evaluated 27 models by providing <question, discharge summaries> pairs directly to the model as input (shown in Figure 11 for both settings). However, why were no additional task instructions provided? Evaluating the models' performance in a Chain-of-Thought (CoT) setting is becoming a de facto method, as it helps to understand the models’ ability to reach answers based on correct rationale and provides greater insights into their performance. Why was CoT or similar methods not used for this task? Is there a specific reason for this choice?
>
> &
>
> In section 4.2, the authors present the findings from Table 3 but do not provide a detailed explanation of the reasons behind these findings. I think a CoT kind of experimental setting (by exploring how models arrive at their answers) could help understand such models' behaviors. Please add some detailed analysis related to these findings.
>
> **A2.**
>
> We appreciate the suggestion to evaluate the models' performance using Chain-of-Thought (CoT) reasoning, as it indeed offers valuable insights into the models’ ability to generate answers. However, applying CoT or similar reasoning methodologies within the clinical domain presents unique challenges that differ significantly from those in the general domain. Specifically, rigorous evaluation of reasoning in clinical contexts necessitates the direct involvement of clinicians to verify the correctness and relevance of the model’s outputs.
>
> Given the limited resources available, particularly in terms of clinician time, we had to prioritize tasks that were most critical to the validation of EHRNoteQA. Instead of conducting a detailed analysis of model performance with CoT reasoning, we focused on two key objectives: (1) ensuring the thorough review and refinement of our dataset by clinicians, and (2) verifying that the LLM scores obtained through our dataset align with the evaluations made by actual clinicians. These tasks were essential to validate the clinical significance and necessity of the dataset itself.
>
> While we acknowledge that an in-depth analysis of the models’ reasoning capabilities would be beneficial for understanding their performance and guiding future model development, such an analysis was beyond the scope of our current study due to time and resource constraints. However, we believe that our focus on the clinical relevance of the dataset is crucial and sufficient for demonstrating its value within the clinical domain.
>
> We agree that exploring models’ reasoning abilities in clinical contexts is an important area for follow-up research, and we consider it a necessary next step to build on our current findings.

---

> > ### Comment · Reviewer_XMWc · 2024-08-27
> >
> > I still believe small analysis with CoT can be very helpful in strengthening the paper.

---

> > > ### Author Rebuttal · Authors · 2024-08-30
> > >
> > > Following your suggestion, we conducted a small-scale analysis using CoT reasoning. We focused on GPT-4, the model that achieved the highest score in the EHRNoteQA, and re-generated responses for the questions it initially got wrong in the free-text format of the EHRNoteQA level 2 data. This time, we included the prompt "Let's think step-by-step" [1] to trigger CoT reasoning, and then re-evaluated the results using the same grading criteria.
> > >
> > > The results showed that using CoT corrected 30% (18 out of 58) of the errors made in the direct generation of free-text responses. For the remaining 40 incorrect responses, we conducted an error analysis with the help of a clinician to examine the reasoning generated by the CoT prompt. The findings for error analysis are summarized as follows:
> > >
> > > 1. **Clinical Reasoning Errors (35%)**: The model often struggled with reasoning within discharge summaries. For example, it misinterpreted the context of medication continuation at discharge or incorrectly specified a final diagnosis that was only suspected but not confirmed.
> > > 2. **Hallucinations in Medical Concepts (27%)**: The model produced incorrect outputs, such as mistakenly classifying Codeine as an antibiotic instead of an opioid analgesic.
> > > 3. **Partial Extraction (18%)**: The model did not always retrieve all relevant medical information from the discharge summaries, omitting key treatment details in some cases.
> > > 4. **Over-Extraction (10%)**: The model sometimes included unnecessary treatments or instructions, such as irrelevant medications for certain conditions.
> > > 5. **Temporal Errors (10%)**: The model occasionally misidentified the timing of events in the patient's medical history, associating lab tests with the wrong hospital visit.
> > >
> > > Although this analysis was limited in scope due to the short rebuttal period and was applied to only one model, we believe that expanding this approach to include more models and scenarios in future research could yield valuable insights.
> > >
> > > [1] Kojima, Takeshi, et al. "Large language models are zero-shot reasoners." Advances in neural information processing systems 35 (2022): 22199-22213.

---

> ### Author Rebuttal · Authors · 2024-08-15
>
> We are grateful for your valuable feedback and suggestions. We have addressed your comments in the following Q-A format (from **Q1** to **Q7**):
>
> **Q1.**
>
> The authors have used GPT-4 to generate <question, answer> pairs based on discharge summaries, including incorrect options. According to the provided statistics, the authors have generated 962 questions for 962 distinct patients, resulting in one question per patient per topic. Why not generate one question per topic for each patient, leading to a dataset with a total of (962 x 8) <question, answer> pairs? This approach could enhance the data size, diversity, and comprehensiveness of the benchmark. Is there a specific reason for the chosen approach of generating only one question per patient in the paper?
>
> **A1.**
>
> We’d like to clarify the rationale behind our data generation approach.
>
> When generating the EHRNoteQA questions using GPT-4, we intentionally did not specify particular topics in the prompt. Instead, we provided only the discharge summaries, allowing the model to generate questions that a clinician might naturally ask based on the information provided. After these questions were generated, they were reviewed and refined by clinicians and then categorized accordingly. Therefore, as you suggested, generating 962*8 questions across eight predefined topics for 962 patients was not feasible within our framework.
>
> We chose not to include specific topics in the prompt for GPT-4 QA generation based on feedback from clinicians who reviewed our initial attempts during the prompt-tuning stage. When specific topics were preassigned, GPT-4 often generated questions that were either unnatural or clinically irrelevant to the patient's case. Our priority was to ensure that each question closely reflects the inquiry a clinician might naturally pose in a real-world setting based on the patient's discharge summaries. Hence, we chose to generate questions without imposing constraints.
>
> Regarding the generation of only one question per patient, this approach was intended to maximize diversity within the limited dataset size. Each QA pair was carefully reviewed and revised by clinicians, which naturally limited the number of questions we could include. To ensure the comprehensiveness of the dataset, we focused on capturing unique patient cases so that our dataset includes a broad range of clinical scenarios.
>
> Thanks to your feedback, we recognize the need to clarify our data generation phase and explain why we adopted the one-question-per-patient approach. We will incorporate a more detailed explanation in Section 3 of the camera-ready version.

---

### Official Review · Reviewer_VCg7 · 2024-07-22
**Review comments**

**Rating:** 8
**Confidence:** 4
**Correctness:** The claims are correct.
**Clarity:** The paper is well written.

**Review:**

My major comments are:

1. How can the diversity of LLM-generated questions be ensured? Will GPT-4 generate similar questions for different patients?
2. The authors should clarify the clinical significance and explain how the proposed dataset can be used or evaluated in future clinical developments of LLMs compared to existing datasets.

**Strengths:**

The paper is well-written and has no major technical flaws. The proposed dataset is clinically significant for LLM applications.

**Additional Feedback:**

NA

**Documentation:**

NA

**Limitations:**

Please refer to my review comments.

**Opportunities For Improvement:**

Please refer to my review comments.

**Relation To Prior Work:**

Related works are discussed.

**Summary And Contributions:**

In this work, the authors propose a QA dataset using the MIMIC-IV dataset. The proposed dataset is generated using GPT-4 and manually reviewed by clinicians. The authors also evaluate the performance of 27 LLMs using this dataset.

---

> ### Author Rebuttal · Authors · 2024-08-15
>
> **Q2.**
> The authors should clarify the clinical significance and explain how the proposed dataset can be used or evaluated in future clinical developments of LLMs compared to existing datasets.
>
> **A2.**
> We would like to highlight the key clinical contributions of the EHRNoteQA dataset:
>
> 1. **QA on Multiple Discharge Summaries**: EHRNoteQA is the first dataset to feature QA pairs based on multiple discharge summaries for a patient. This sets it apart from existing discharge summary QA benchmarks, which typically rely on a single discharge summary per patient.
> 2. **Diversity of Question Categories**: Our dataset covers a wide range of question categories—8 in total—which is broader than that of other benchmarks. This diversity enhances the comprehensiveness of model evaluation by enabling evaluation in more diverse clinical aspects (i.e., treatment, assessment, problem, etiology, sign/symptom, vitals, test results, history, instruction, plan).
> 3. **Alignment with Clinician Evaluation**: A critical advantage of using EHRNoteQA lies in its alignment with direct clinician evaluations, ensuring that the performance of LLMs is closely tied to real-world clinical relevance compared to other benchmark datasets.
>
> Regarding the utility of our dataset in future clinical developments of LLMs, our experiment results (see Table 4) shows that evaluations using EHRNoteQA exhibit higher correlation with clinician evaluations compared to using other benchmark datasets, including discharge summary QA datasets (emrQA, Yue et al.), general clinical benchmarks (MedQA, PubMEdQA, MMLU (medical), MedMCQA), and general-purpose benchmarks (ARC, HellaSwag, MMLU, TruthfulQA, Winogrande, WSM8K). This higher alignment indicates that EHRNoteQA is the most relevant benchmark for assessing and selecting LLMs for real-world clinical deployment, particularly in the context of QA on patient discharge summaries.

---

> ### Author Rebuttal · Authors · 2024-08-15
>
> Thank you for your appreciation and valuable evaluation of our work. Please kindly find the reviews below.
>
>
> **Q1.**
> How can the diversity of LLM-generated questions be ensured? Will GPT-4 generate similar questions for different patients?
>
> **A1.**
> As you pointed out, the diversity in LLM-generated questions cannot be guaranteed, and similar questions can be generated for different patients. However, we want to highlight that EHRNoteQA exhibits greater diversity, both semantically and syntactically, compared to existing discharge summary QA datasets.
>
> Semantically, our dataset covers a broad range of 8 topics (i.e., treatment, assessment, problem, etiology, sign/symptom, vitals, test results, history, instruction, plan). In contrast, existing discharge summary QA datasets such as [1] emrQA and [2] Yue et al. mostly consist of questions with limited topics, such as topics derived from i2b2 annotations (e.g., smoking, medication).
>
> To quantify syntactic diversity,  we analyzed the percentage of unique bi-grams across different discharge summary QA datasets. As presented in the table below, EHRNoteQA shows a higher percentage of unique bi-grams, indicating greater syntactic diversity.
>
> |                   | emrQA | CliniQG4QA | EHRNoteQA |
> |-------------------|-------|------------|-----------|
> | Unique Bi-Grams (%) | 16%   | 24%        | 27%       |
>
>
> Regarding the concern about similar questions being generated for different patients, this indeed reflects real-world clinical practice. It is common for similar questions to be asked on different patients' discharge summaries (e.g., What was the reason for the patient's admission and what treatment was the patient provided during their stay?). Additionally, from a model evaluation perspective, even if the model answers a question correctly for one patient, it does not guarantee the model's ability to answer a similar question for another patient, given the distinct nature of each patient’s discharge summaries (similar questions for different patients may have significantly different answers).
>
> [1] Pampari, A., Raghavan, P., Liang, J. and Peng, J., 2018. emrqa: A large corpus for question answering on electronic medical records. arXiv preprint arXiv:1809.00732.
>
> [2] Yue, X., Zhang, X.F., Yao, Z., Lin, S. and Sun, H., 2021, December. Cliniqg4qa: Generating diverse questions for domain adaptation of clinical question answering. In 2021 IEEE International Conference on Bioinformatics and Biomedicine (BIBM) (pp. 580-587). IEEE.

---

> ### Comment · Reviewer_VCg7 · 2024-08-19
> **Response to author rebuttal**
>
> Thanks to the authors for their response. My questions have been addressed. I hope the author can revise these changes in their final version.

---

### Official Review · Reviewer_2Hp9 · 2024-07-24
**A new medical QA benchmark, "EHRNoteQA", for multi-document QA over discharge summaries**

**Rating:** 7
**Confidence:** 3
**Correctness:** No issues with correctness on my read.
**Clarity:** I found the paper quite pleasant to r…

**Review:**

Unclear why this section exists as it's covered by all the sections above/below so skipping!

**Strengths:**

- The questions, while synthetically generate by GPT-4, are manually verified and corrected by domain experts
- The evaluation is quite comprehensive, especially with respect to the # of LLMs evaluated
- The improvement in correlation between results on this benchmark and human judgment compared to existing medical QA benchmarks is genuinely exciting

**Additional Feedback:**

N/A.

**Documentation:**

Really would have liked to see a GitHub link in the main paper, pointing to a repo containing thorough documentation about how to use the dataset. Instead I see in the appendix: "The dataset will be released via GitHub upon publication." I feel like this is not good enough for a D&B paper.

Also, for accessing the dataset a Google Drive link is provided in the appendix, with the comment: "Note that this link will be deprecated after the review process ends". Why deprecated? Hopefully because a different and possibly better way will be used to share the data?

**Ethics:**

No significant ethical concerns.

**Limitations:**

- The dataset is heavily skewed towards "Treatment" style of questions (e.g. "What was the treatment provided for the patient’s left breast cellulitis?"). Do the authors want to comment on any implications of this in the Limitations section?
- If I am reading 3.1 correctly, no discharge summary context exceeds 7000 tokens. Therefore, the benchmark does not test for more extreme cases outside this range (something like 30% of cases in MIMIC-IV if I am reading line 126 correctly). Maybe a note should be added on this in the limitations?

**Opportunities For Improvement:**

- Usage of the dataset doesn't look terribly difficult, but could be made easier by hosting it one or more of the standard platforms our community uses, like the HuggingFace Hub, assuming there is no issues with sharing MIMIC data this way. Even if there is, there should be ways around that (e.g. only share IDs + questions + answers and provide easy to use code for someone with access to MIMIC to build the full benchmark). Would also love to see clear documentation and easy to use code around the evaluation process.
- It's unclear to me why GPT-4 would be needed for the multiple-choice question answer setup. Is this strictly because a language model may provide an invalid answer, or a non-exact match? In any case, it would make it difficult for others to mimic your evaluation as a you mention you need to use an instance of GPT-4 on Azure’s HIPAA-compliant platform to adhere to the MIMIC-IV data usage regulations. Maybe some further guidance here on how to evaluate in the MCQA setup when this isn't possible would be a good addition to the paper.

**Relation To Prior Work:**

Clearly discussed in Section 2.

**Summary And Contributions:**

Introduces a new medical QA benchmark, "EHRNoteQA", for multi-document QA over discharge summaries. The benchmark is based on MIMIC-IV EHR, the questions are generated by GPT-4 and then manually reviewed by 3 clinicians. For each question there is 1 correct answer and 4 distractors. The benchmark supports open QA and multiple-choice QA. The authors demonstrate that performance on this benchmark better correlates with human judgment than existing medical QA benchmarks.

---

> ### Author Rebuttal · Authors · 2024-08-15
>
> Thank you for your careful review of our paper.
>
> We would be grateful if you could let us know if our response addressed your concerns or if there are any further questions or feedback.

---

> > ### Comment · Reviewer_2Hp9 · 2024-08-19
> >
> > All my comments are addressed!

---

> ### Author Rebuttal · Authors · 2024-08-15
>
> **Q4.**
> If I am reading 3.1 correctly, no discharge summary context exceeds 7000 tokens. Therefore, the benchmark does not test for more extreme cases outside this range (something like 30% of cases in MIMIC-IV if I am reading line 126 correctly). Maybe a note should be added on this in the limitations?
>
> **A4.**
> The decision to limit discharge summary contexts to 7000 tokens was made to align with the current capabilities and limitations of existing models, specifically reflecting the maximum context length that current models can handle. As models evolve to handle longer contexts, we plan to extend our dataset to include cases with more admissions and longer discharge summaries, as mentioned in our future work section. As you noted, however, this choice does exclude cases with longer context, which may restrict the evaluation of more extreme cases. Based on your feedback, we will move this to the limitations section in the camera-ready version.

---

> ### Author Rebuttal · Authors · 2024-08-15
>
> **Q3.**
> The dataset is heavily skewed towards "Treatment" style of questions (e.g. "What was the treatment provided for the patient’s left breast cellulitis?"). Do the authors want to comment on any implications of this in the Limitations section?
>
> **A3.**
> We appreciate the opportunity to clarify the high proportion of treatment-related questions of our work. Firstly, we would like to highlight that the question categories in our dataset are not mutually exclusive; a single question can belong to multiple categories, as indicated in the captions of Table 2. For example, the question “What were the primary causes of the patient’s shortness of breath and how were these managed during the hospital stay?” is classified under both etiology and treatment. The higher frequency of treatment-related questions can be attributed to the fact that treatment questions often are asked along with other categories, such as etiology, problem, sign/symptom, test results.
>
> We consulted with clinicians who participated in our dataset construction, and confirmed that treatment-related questions are indeed frequently asked along with other types of questions in clinical practice. Below are some examples in EHRNoteQA of treatment questions paired along with other categories.
>
> - **Treatment & Problem**: What was the primary injury and procedure used to treat it from the accident?
> - **Treatment & Sign/Symptom**: What were the symptoms the patient presented with in the second admission and did any new medications were added to the list from the first admission?
> - **Treatment & Test Results**: What was the result of the blood culture test conducted on admission and what was the corresponding antibiotics prescribed?
>
> In light of your comments, we will expand on this explanation in Appendix D of our paper in the camera-ready version, providing additional examples of questions that span multiple categories.

---

> ### Author Rebuttal · Authors · 2024-08-15
>
> **Q2.**
> It's unclear to me why GPT-4 would be needed for the multiple-choice question answer setup. Is this strictly because a language model may provide an invalid answer, or a non-exact match? In any case, it would make it difficult for others to mimic your evaluation as you mention you need to use an instance of GPT-4 on Azure’s HIPAA-compliant platform to adhere to the MIMIC-IV data usage regulations. Maybe some further guidance here on how to evaluate in the MCQA setup when this isn't possible would be a good addition to the paper.
>
> **A2.**
> We acknowledge that using regular expressions or similar matching techniques can make the multiple-choice evaluation more accessible to others. However, the choice to employ GPT-4 was driven by the need for precise multiple-choice evaluation due to the variability in model outputs under zero-shot settings, where the model outputs can vary significantly. For example, these outputs may range from explicit answers like "The answer is A." or simply “A,” to more implicit responses such as "The patient underwent pipeline embolization," where the answer choice letter is not directly provided. Additionally, some outputs may include reasoning (e.g., "The choice B is not the answer because..."), further complicating the evaluation process with fixed-rules.
>
> As models with longer context lengths become available in the future, providing few-shot examples, which can enable more controlled outputs, may facilitate rule-based evaluations such as using regular expressions or similar matching. Unfortunately, this is currently not an option.
>
> For those unable to use GPT-4 on a HIPAA-compliant platform, we suggest an alternative approach using evaluation on probabilities of the answer index (e.g., A, B, C, D, E). Although this method may not be as optimal as GPT-4, it can still serve as a reasonable alternative for scoring, as demonstrated by the correlation comparisons in Table 4. We do, however, advise caution with this alternative, as our experiments show that it may not be as reliable as GPT-4.

---

> ### Author Rebuttal · Authors · 2024-08-15
>
> Thank you for the time and effort to review our work and provide valuable comments. Please kindly find the reviews below.
>
> **Q1.**
> Usage of the dataset doesn't look terribly difficult, but could be made easier by hosting it one or more of the standard platforms our community uses, like the HuggingFace Hub, assuming there is no issues with sharing MIMIC data this way. Even if there is, there should be ways around that (e.g. only share IDs + questions + answers and provide easy to use code for someone with access to MIMIC to build the full benchmark). Would also love to see clear documentation and easy to use code around the evaluation process.
>
> &
>
> Really would have liked to see a GitHub link in the main paper, pointing to a repo containing thorough documentation about how to use the dataset. Instead I see in the appendix: "The dataset will be released via GitHub upon publication." I feel like this is not good enough for a D&B paper.
> Also, for accessing the dataset a Google Drive link is provided in the appendix, with the comment: "Note that this link will be deprecated after the review process ends". Why deprecated? Hopefully because a different and possibly better way will be used to share the data?
>
> **A1.**
> We completely agree that hosting our dataset on platforms like HuggingFace Hub would significantly enhance accessibility. However, the MIMIC-IV dataset usage agreement imposes strict guidelines (https://physionet.org/news/post/mimic-derived-datasets-models), specifically that “Any derived datasets or models should be treated as containing sensitive information. If you wish to share these resources, they should be shared on PhysioNet under the same agreement as the source data.” Consequently, sharing a portion of the data, such as patient IDs, questions, answers, on public platforms is unfortunately not permitted. The Google Drive link was provided solely for review purposes and will be deprecated as the dataset must be hosted on PhysioNet.
>
> In response to your concern, we share the EHRNoteQA dataset link available on Physionet: https://doi.org/10.13026/acga-ht95. The dataset can be accessed by users with Physionet credentials. We have also prepared a GitHub repository that includes detailed documentation and code in the following link: https://github.com/ji-youn-kim/EHRNoteQA. The repository includes scripts for 1) merging EHRNoteQA with the MIMIC-IV discharge summaries, followed by preprocessing the MIMIC-IV discharge summaries (e.g., eliminating redundant whitespaces), 2) generating model outputs, 3) evaluating model outputs. In the documentation (i.e., README.md file in the GitHub repository), we provide step-by-step instructions on package installation, data preprocessing, model output generation, and evaluation, ensuring ease of use.
>
> Addressing your feedback, we will make sure to include links to both the EHRNoteQA dataset on Physionet and the GitHub repository in the abstract section of the camera-ready version of our paper. Furthermore, should a collaboration between HuggingFace and Physionet become viable in the future, we will promptly make the dataset available on HuggingFace as well.

---

### Official Review · Reviewer_gAX8 · 2024-08-02
**Benchmarking LLMs for question answering over clinical discharge summaries**

**Rating:** 9
**Confidence:** 4
**Correctness:** Yes
**Clarity:** Yes

**Review:**

This paper builds a significant benchmark to test LLMs for deployment to clinical settings.  The details of the benchmark construction and evaluation are presented well. The involvement of clinical experts has been documented, which makes the benchmark more authentic and closer to real-world settings.

**Strengths:**

1. The benchmark has significance in AI for Healthcare domain. It will help in rigorously testing LLMs in close to real world scenarios and quicken their use from PoCs to deployment.
2. The benchmark is made more authentic because the synthetic dataset is analysed, edited and validated by clinicians.
3. The

**Additional Feedback:**

Please refer to Opportunities for improvement section.

**Documentation:**

Yes, more details in the appendix in the supplementary attachment.

**Limitations:**

Yes

**Opportunities For Improvement:**

Perhaps the manual evaluation techniques and metrics can also be used like Bleu, Rouge, semantic similarity Exact match. It could be worth comparing them to LLM based evaluation.

**Relation To Prior Work:**

Yes

**Summary And Contributions:**

This paper proposes a benchmark for LLMs for question answering over clinical discharge summaries, noting the existing challenges for medical professional and ML models to synthesise information across multiple and lengthy discharge summaries. The QA pairs are synthetically generated using GPT-4 but verified by clinical experts for clinical relevance. Dataset reflects a diverse set of 8 topics. The answer is expected in two format - open ended and multiple choice. Both formats have evaluation defined. 27 LLMs are evaluated and benchmarked on this dataset.

This contribution is significant as the benchmark is constructed as close as possible to a real-world scenario, enabling rigorous testing and speeding deployment of LLMs in clinical settings.

---

> ### Author Rebuttal · Authors · 2024-08-15
>
> Thank you for your appreciation and valuable evaluation of our work. Please kindly find the reviews below.
>
> **Q1.**
> Perhaps the manual evaluation techniques and metrics can also be used like Bleu, Rouge, semantic similarity Exact match. It could be worth comparing them to LLM based evaluation.
>
> **A1.**
> In our study on EHRNoteQA, we employed three methods for evaluating models on open-ended tasks: 1) GPT-4 evaluation, 2) BLEU, and 3) ROUGE-L. The results of these evaluations are presented in Supplementary Material (Appendix) Table 14, and the alignment between LLM scores obtained through above methods and the clinician-evaluated LLM scores is detailed in Table 4 of the main paper.
>
> Based on your feedback, we have also incorporated **exact match scores** and **semantic similarity** into our evaluation metrics. For Semantic Similarity, we utilized two models: sentenceBERT and clinicalBERT, measuring the cosine similarity between the embeddings of the model output and the gold standard answers. The results of these additional evaluations are presented below.
>
> | Model                           | EM (F1 score) | Cosine (SentenceBERT) | Cosine (Clinical BERT) |
> |---------------------------------|---------------|-----------------------|------------------------|
> | Llama2-70b-chat                 | 0.2777        | 0.6840                | 0.9321                 |
> | qCammel-70                      | 0.2684        | 0.6804                | 0.9238                 |
> | Camel-Platypus2-70b             | 0.2851        | 0.6878                | 0.9285                 |
> | Platypus2-70b-Instruct          | 0.3177        | 0.6944                | 0.9364                 |
> | MPT-30b-Instruct                | 0.3016        | 0.6357                | 0.9125                 |
> | Llama2-13b-chat                 | 0.2596        | 0.6674                | 0.9281                 |
> | Vicuna-13b                      | 0.2803        | 0.6679                | 0.9278                 |
> | WizardLM-13b                    | 0.2536        | 0.6679                | 0.9273                 |
> | qCammel-13                      | 0.2495        | 0.6624                | 0.9205                 |
> | OpenOrca-Platypus2-13b          | 0.2702        | 0.6679                | 0.9198                 |
> | Camel-Platypus2-13b             | 0.2302        | 0.6535                | 0.9167                 |
> | Synthia-13b                     | 0.3184        | 0.6833                | 0.9341                 |
> | Llama2-7b-chat                  | 0.2452        | 0.6634                | 0.9244                 |
> | Vicuna-7b                       | 0.2684        | 0.6599                | 0.9284                 |
> | Mistral-7b-Instruct             | 0.3072        | 0.6829                | 0.9357                 |
> | MPT-7b-8k-instruct              | 0.2979        | 0.6280                | 0.9174                 |
> | Dolphin-2.0-mistral-7b          | 0.3055        | 0.6637                | 0.9301                 |
> | Mistral-7b-OpenOrca             | 0.3280        | 0.6991                | 0.9399                 |
> | SynthIA-7b                      | 0.3021        | 0.6856                | 0.9336                 |
>
>
>
> When we measured the correlation between these additional evaluation results and the clinician-evaluated LLM scores, we observed the following outcomes:
>
>
>
> |  | Clinician A |  | Clinician B |  | Clinician C |  |
> |--------|-------------|-------------|-------------|-------------|-------------|-------------|
> |  | Spearman | Kendall | Spearman | Kendall | Spearman | Kendall |
> | **GPT-4**        | **0.770**       | **0.609**    | **0.805**       | **0.617**    | **0.801**       | **0.657**    |
> | **BLEU**         | 0.155       | 0.112    | 0.037       | 0.059    | 0.014       | -0.006   |
> | **ROUGE-L**      | 0.500       | 0.324    | 0.398       | 0.283    | 0.356       | 0.241    |
> | **EM (F1 score)**| 0.422       | 0.288    | 0.336       | 0.236    | 0.266       | 0.194    |
> | **Cosine (SentenceBERT)** | 0.710  | 0.524    | 0.726       | 0.555    | 0.652       | 0.453    |
> | **Cosine (Clinical BERT)** | 0.536  | 0.382    | 0.552       | 0.389    | 0.394       | 0.288    |
>
>
> While using GPT-4 still exhibited the highest correlation with clinician evaluations, the semantic similarity method—particularly using sentenceBERT for cosine similarity—demonstrated the next highest correlation. This suggests that the sentenceBERT-based cosine similarity method could serve as a viable alternative evaluation metric for those who are limited to use GPT-4.

---

### Decision · Program_Chairs · 2024-09-26

**Decision:**

Accept (Poster)

**Comment:**

This paper presents a new medical QA benchmark, EHRNoteQA, designed for multi-document question answering over discharge summaries. The benchmark is built on the MIMIC-IV EHR dataset, with questions generated by GPT-4 and manually reviewed by three clinicians. For each question, there is one correct answer and four distractors. The benchmark supports both open QA and multiple-choice QA formats. The authors demonstrate that performance on this benchmark correlates more closely with human judgment compared to existing medical QA benchmarks. While all reviewers recognize the benchmark's significance in advancing AI for healthcare, the authors should provide additional explanations regarding evaluation methods, data sharing, question diversity, etc. Overall, this is a high-quality benchmark paper, and I recommend accepting it.